# MMSᴇᴀʀᴄʜ-Pʟᴜs:
# Bᴇɴᴄʜᴍᴀʀᴋɪɴɢ Pʀᴏᴠᴇɴᴀɴᴄᴇ-Aᴡᴀʀᴇ Sᴇᴀʀᴄʜ ғᴏʀ Mᴜʟᴛɪᴍᴏᴅᴀʟ Bʀᴏᴡsɪɴɢ Aɢᴇɴᴛs

**Xijia Tao[1*], Yihua Teng[2*], Xinxing Su[2*], Xinyu Fu[2], Jihao Wu[2], Chaofan Tao[2],**
**Ziru Liu[2], Haoli Bai[2], Rui Liu[2†], Lingpeng Kong[1†]**
[1]The University of Hong Kong (HKU)    [2]Huawei Inc.

## Abstract

Existing multimodal browsing benchmarks often fail to require genuine multimodal reasoning, as many tasks can be solved with text-only heuristics without vision-in-the-loop verification. We introduce MMSearch-Plus, a 311-task benchmark that enforces multimodal understanding by requiring extraction and propagation of fine-grained visual cues through iterative image–text retrieval and cross-validation under retrieval noise. Our curation procedure seeds questions whose answers require extrapolating from spatial cues and temporal traces to out-of-image facts such as events, dates, and venues. Beyond the dataset, we provide a model-agnostic agent framework with standard browsing tools and a set-of-mark (SoM) module, which lets the agent place marks, crop subregions, and launch targeted image/text searches. SoM enables provenance-aware zoom-and-retrieve and improves robustness in multi-step reasoning. We evaluated closed- and open-source MLLMs in this framework. The strongest system achieves an end-to-end accuracy of 36.0%, and integrating SoM produces consistent gains in multiple settings, with improvements up to +3.9 points. From failure analysis, we observe recurring errors in locating relevant webpages and distinguishing between visually similar events. These results underscore the challenges of real-world multimodal search and establish MMSearch-Plus as a rigorous benchmark for advancing agentic MLLMs.

## 1 Iɴᴛʀᴏᴅᴜᴄᴛɪᴏɴ

Large multimodal language models (MLLMs) increasingly act as agents that integrate vision, language, and web search to answer information-seeking questions. Several recent benchmarks study this capability by pairing images with web browsing or image search tools. However, despite their multimodal interface, existing multimodal browsing benchmarks rarely demand sustained, fine-grained visual reasoning or long-horizon, tool-augmented search.

This paper introduces **MMSearch-Plus**, a new benchmark that closes this gap. MMSearch-Plus is designed to require (i) *localized, exhaustive visual reasoning* over subtle cues, (ii) *robust verification* under noisy or conflicting retrieval, and (iii) *multi-step tool use* that interleaves text and image search with region-level visual analysis. To support this, we contribute:

- **A challenging multimodal browsing benchmark** that matches the long-horizon difficulty of text-based browsing tasks (e.g., BrowseComp) while adding visual reasoning requirements that cannot be bypassed by a single strong image search.

- **Spatial–Temporal Extrapolation**, a principled curation procedure that constructs hard questions by requiring models to extrapolate from localized spatial cues and temporal signals to out-of-image facts such as dates, events, or locations.

- **A general, model-agnostic agent framework** that interleaves text/image search with a *Set-of-Mark* (SoM) zoom-in pathway, enabling region-seeded retrieval (`zoom_in`/`image_search`) and systematic analysis of how cropping and "thinking with images" affect current MLLMs.

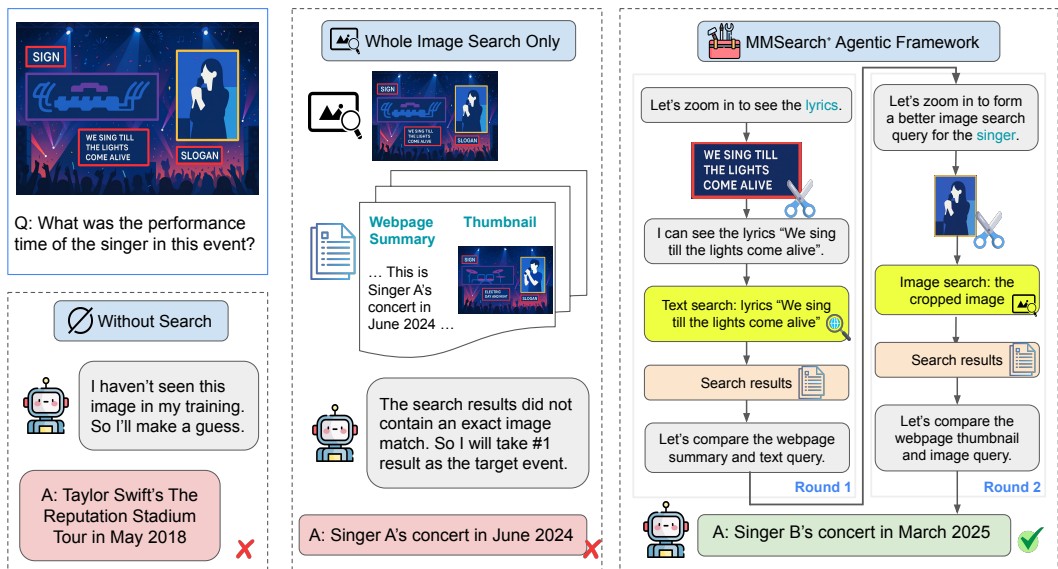

Figure 1: Three multimodal reasoning paradigms: 1) *Without search*: an MLLM answers a visual question using internal knowledge only; 2) *Whole-image search*: an MLLM leverages external search results retrieved from the full VQA image; 3) *MMSearch-Plus agentic framework*: an MLLM freely calls visual tools (cropping, OCR) and search engines to extract fine-grained cues and perform precision search. A real example is shown in Figure 8.

We next motivate why such a benchmark is needed. Multimodal browsing benchmarks such as MM-Search (Jiang et al., 2024) primarily test an MLLM's ability to coordinate text and images, but many of their questions admit relatively fixed workflows. A strong image search engine often retrieves pages whose surrounding text directly contains the answer, allowing unimodal LLMs—without visual perception—to perform competitively. In these cases, multimodality collapses to a narrow form of *image–source cross-validation*, and fine-grained visual reasoning plays only a minor role.

In contrast, recent text-only browsing tasks such as BrowseComp (Wei et al., 2025) emphasize persistence, multi-step evidence gathering, and complex search strategies. State-of-the-art MLLMs score below 2% with browsing tools, indicating substantial headroom and revealing a discrepancy: multimodal browsing benchmarks are far easier than their text-only counterparts, even though real-world multimodal tasks often require deeper reasoning.

**Design principles for a challenging multimodal browsing benchmark.** Difficult web-based visual tasks typically exhibit three recurring challenges:

1. **Noisy or conflicting retrieval.** Image search may surface near-duplicates, irrelevant scenes, or contradictory candidates; solving the task requires cross-source consistency checks and multimodal verification.

2. **Exhaustive, part-based visual reasoning.** When holistic matches fail, agents must reason over subregions—cropping, zooming, and re-searching—to accumulate fragmented spatial cues.

3. **Long, tool-augmented reasoning chains.** Realistic tasks require sustained coordination of visual tools (cropping, OCR), textual browsing, and multimodal evidence integration.

Benchmarks such as MMSearch do not consistently elicit these behaviors: many images contain a single dominant entity that quickly anchors a search, and the subsequent steps are primarily text-based.

**MMSearch-Plus.** To construct tasks that inherently require fine-grained visual reasoning, MMSearch-Plus introduces *Spatial–Temporal Extrapolation*: we curate images containing multiple subtle signals—micro-text, uniform variations, broadcast overlays, layout signatures—that jointly

imply facts not explicitly present in the prompt. Models must extract these cues, propagate them through iterative searches, and verify provenance amid retrieval noise. This better reflects real-world scenarios where neither the prompt nor the image alone suffices; for instance, determining the date of an esports broadcast or identifying a specific match from nearly identical sports frames. Finally, we develop a general agent framework with *Set-of-Mark* (SoM) to study how region-level visual grounding influences browsing agents. SoM provides human-verified marks and region-seeded queries, enabling "thinking with images" through structured zoom-in reasoning. Ablations quantify how cropping and localized visual extraction affect current MLLMs.

Together, MMSearch-Plus provides a high-difficulty, multimodal benchmark and analysis toolkit for evaluating the next generation of multimodal browsing agents. It differs from concurrent efforts such as BrowseComp-VL (Geng et al., 2025) and MM-BrowseComp (Li et al., 2025b) in both data sources and its agent framework, which emphasizes sustained fine-grained visual reasoning.

## 2 METHOD

### 2.1 SPATIAL-TEMPORAL EXTRAPOLATION

**Summary.** We test an agent's ability to infer facts that are *not* literally present in the pixels or the instant shown, requiring it to recover the broader event by combining partial visual cues with retrieval and reasoning across time and space. This emphasis discourages purely perceptual shortcutting and directly measures uncertainty-aware planning, precise event localization, and evidence-backed browsing.

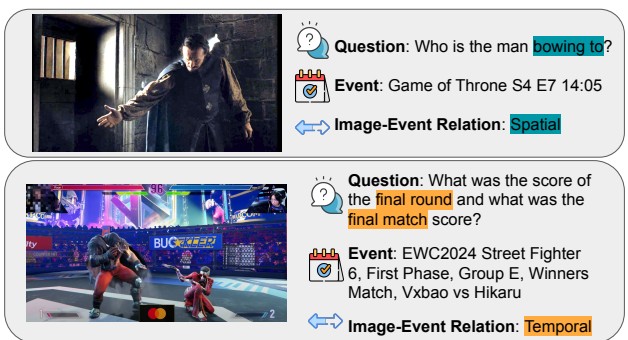

Figure 2: Examples of spatial and temporal relations for question curation.

A central challenge in BrowseComp-like benchmarks is the ballooning intermediate search space induced by soft, fuzzy constraints, which demands non-trivial cross-validation to identify the correct target. Rather than remixing text-only corpora, we construct multimodal problems that *naturally* expand this search space: from a handful of visual fragments tied to real-world events (in the spirit of GeoGuessr AB (2013)), the agent must hypothesize the underlying source event and verify it against retrieved evidence. Difficulty is modulated by the richness of visual and textual cues; even a single crop can widen the candidate set dramatically, eliciting iterative hypothesis–test–refine behavior and long trajectories that require robust context tracking.

Once the source event is pinned down, our questions probe metadata and support multi-hop reasoning. To elevate difficulty, we introduce *Spatial-Temporal Extrapolation*: instead of querying what is directly visible, we ask what is contextually implied but physically absent, compelling reasoning beyond the frame and moment to reconstruct the broader situation. **Spatial** extrapolation targets unseen entities (e.g., off-screen participants, individuals facing away, occluded signage), while **temporal** extrapolation probes events immediately before or after the depicted moment (e.g., lineups, the next play/goal, or the subsequent segment/episode). Solving these requires agents to (i) precisely localize the event (time, match, or episode), and (ii) retrieve and integrate wider contextual knowledge from diverse sources to produce an evidence-grounded answer.

### 2.2 DATA COLLECTION

Building on the Spatial–Temporal Extrapolation framework, we instantiated the data collection process as follows.

We began with a set of predefined categories and generated search keywords using a mix of LLM assistance and manual curation. Data were sourced from publicly accessible video-sharing platforms and open-access scholarly repositories (i.e., arXiv). Queries for consumer video content (e.g., *Vlog*)

were issued to video platforms, while those in the *Acad.* category were issued to arXiv to retrieve relevant research papers (PDFs). All materials are used only for research. We also mask sensitive content (e.g., faces and vehicle license plates). In our pipeline, both videos and papers are treated as *events*.

For videos, human annotators manually selected keyframes suitable for constructing questions. The keyframes underwent anonymization for sensitive content. For papers, annotators extracted figures or tables, capturing them via screenshots. A screenshot was considered suitable if it (i) contained vague or noisy information that challenged perception, (ii) depicted less familiar entities requiring external retrieval to answer associated questions, and (iii) could not be searched with Google image search directly. This ensures that most, if not all, screenshots are not in the training dataset, and the essential information for answering the question is neither in the training corpora nor directly retrievable via a direct image search.

**Filtering.**  Although the collection focused on new or rarely seen events, we observed that certain closed-source models (e.g., GPT-4o, GPT-5, Gemini-2.5-Pro) could occasionally solve questions without external search. To counteract this, we employed an adversarial filtering procedure to increase difficulty. Specifically, annotators removed or obfuscated questions answerable without retrieval through the following methods:

- **Cross-validation:** Annotators designed problems outside their own knowledge base, then verified solvability by testing on at least two closed-source MLLMs.
- **Image perturbation:** Critical visual subregions were blurred or masked.
- **Iterative refinement:** Questions that remained trivially solvable were discarded.

**Dataset updates.**  Some problems that originally required retrieval later became solvable without search because newer closed-source MLLMs began to recognize previously unfamiliar entities or figures. This temporal drift is unavoidable as models continue to incorporate more recent data. We therefore commit to regularly refreshing the benchmark to suppress such internal-knowledge shortcuts.

## 2.3 DATASET STATISTICS

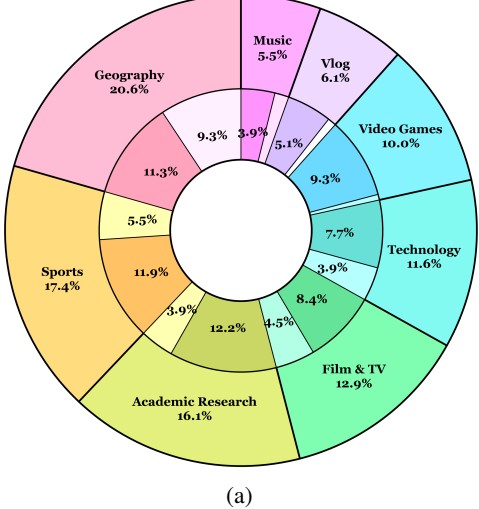

| Statistic | Number |
|---|---|
| Total questions w/ images | 311 |
| Primary categories | 8 |
| Secondary categories | 43 |
| Unique images | 441 |
| Unique questions | 279 |
| Unique answers | 306 |
| Maximum question length | 42 |
| Maximum answer length | 20 |
| Average question length | 12.5 |
| Average answer length | 3.7 |
| Difficulty distribution | |
| - Easy | 94 (30.2%) |
| - Difficult | 217 (69.8%) |

(a)

(b)

Figure 3: (a) Distribution of primary categories in MMSearch-Plus, with light and dark inner sectors representing the easy and difficult splits within each category. (b) Summary of statistics.

Figure 3 depicts the distribution of MMSearch-Plus tasks across eight primary categories, which is approximately balanced. Each task includes at least one image, with additional images provided when a single view is insufficient. The benchmark primarily comprises short-answer questions,

with answers averaging 3.7 words. In hindsight, we labeled the difficulty level of a task as *Easy* if it can be answered correctly without search or with only image search by o3 or Gemini-2.5-Pro. Otherwise, the task is labeled as *Difficult* (interchangeably *Hard*).

## 2.4 BENCHMARK METRIC AND EVALUATION

We adopt accuracy as the primary evaluation metric. For each question, a set of acceptable answers is provided. We use LLM-as-a-judge to compare model predictions against this set, deeming an answer correct if it matches any acceptable variant. Validation through manual inspection showed complete agreement between human judgment and the assessment of GPT-4o.

## 2.5 SEARCH AGENT FRAMEWORK

**Overview.** We implement a model-agnostic web agent that interleaves *text search*, *image search*, and a *zoom-in* tool. Ranked results are obtained via SerpApi; the surrounding framework (i.e., the system that includes the MLLM agent) performs lightweight post-processing to summarize pages, relate images to the query, and manage multi-round context. To reduce latency and token cost across repeated steps/evaluations, the framework caches image-search returns (ranked URLs/thumbnails) and their MLLM summaries per benchmark image. The framework maintains a threaded conversation state (tool calls, cropped views, summaries, and hypotheses) to support long-horizon, contextualized reasoning. It is compatible with sub-image cropping and sub-image search for fine-grained analysis.

**Text and image search.** At any step, the agent may invoke *text* or *image* search. SerpAPI returns ranked candidates, from which the framework forwards a small set of top results to the model (images are summarized and cached; webpages are summarized). Retrieved webpages are summarized by an MLLM (Gemini in our runs) into two fields:

- `web_info`: task-conditioned *semantic* summaries of page content that may support answering the current question;
- `related_info`: evidence linking a result's thumbnail/lead image(s) to the query image (e.g., matching signage, layouts, or micro-text).

These summaries are designed to compress interaction history for use within limited MLLM context windows; metadata such as URL/title is readily handled by SerpAPI. All retrieved and generated artifacts are integrated into the conversation state to enable contextualized evaluation across turns.

**Zoom-in via Set-of-Mark (SoM).** To enable precise, provenance-aware inspection, we augment the agent with a *Set-of-Mark (SoM)* module (Yang et al., 2023). In this work, we refer to SoM as a list of human-verified bounding boxes per task image (examples in Fig. 9). In the first turn, we present both the raw image and an overlaid version with box outlines and indices, avoiding occlusion while allowing cross-reference.

## 3 EXPERIMENT

**Evaluated Models.** We evaluate both closed-source and open-source MLLMs on our benchmark. Closed-source models include **Gemini-2.5-Pro** (hereafter *Gemini*) (Team & DeepMind, 2025), **o3** (OpenAI, 2025b) and **GPT-5** (OpenAI, 2025a), representing the current frontier of proprietary multimodal systems. On the open-source side, we focus on **Qwen-2.5-VL-72B-Instruct** (hereafter *Qwen*) (Bai et al., 2025). Expert human annotators were instructed to work with a Chrome browser for at least 10 minutes for each task, and capped at 20 minutes.

**Implementation Details.** To ensure a fair comparison, we standardize the search-and-reasoning pipeline across all models. We consider five search modes:

(a) **Without Search.** The MLLM answers using only the question and image inputs.

Table 1: End-to-end results on the MMSEARCH$^+$ benchmark, across search modes. All numbers are accuracy (%). Darker blue means higher accuracy. The best and second-best methods under each setting are shown in bold and underline, respectively.

| Model / Search Mode | Avg | By Category | | | | | | | | Difficulty | |
|---|---|---|---|---|---|---|---|---|---|---|---|
| | | Geo. | Sports | Acad. | Film/TV | Tech | Games | Vlog | Music | Easy | Hard |
| **Human** | | | | | | | | | | | |
| Browser | 22.8 | 20.3 | 25.9 | 20.0 | 25.0 | 19.4 | 16.1 | 31.6 | **35.3** | 34.0 | 18.0 |
| **Closed-source** | | | | | | | | | | | |
| o3 (2025-04-16) | | | | | | | | | | | |
| Without Search | 15.1 | 31.2 | 14.8 | 6.0 | 17.5 | 13.9 | 3.2 | 5.3 | 11.8 | 50.0 | 0.0 |
| Image Search | 19.3 | 28.1 | 14.8 | 18.0 | 30.0 | 22.2 | 3.2 | 5.3 | 17.6 | **63.8** | 0.0 |
| Text Search | 37.0 | 43.8 | **35.2** | 48.0 | 30.0 | 44.4 | 16.1 | 31.6 | 29.4 | 50.0 | **31.3** |
| Full Rollout | 36.0 | 35.9 | 24.1 | **50.0** | 42.5 | 44.4 | 16.1 | **42.1** | 29.4 | 54.3 | 28.1 |
| Full Rollout + SoM | 37.6 | **45.3** | 29.6 | 46.0 | 45.0 | **45.7** | 16.1 | 26.3 | 29.4 | 62.8 | 26.9 |
| GPT-5 | | | | | | | | | | | |
| Without Search | 10.3 | 21.9 | 7.4 | 4.0 | 7.5 | 8.3 | 0.0 | 5.3 | 15.8 | 27.7 | 2.8 |
| Image Search | 16.4 | 25.0 | 11.1 | 14.0 | 22.5 | 19.4 | 3.2 | 0.0 | 29.4 | 50.0 | 1.8 |
| Full Rollout + SoM | 35.4 | 35.9 | 27.8 | 48.0 | **50.0** | 41.7 | 6.5 | 36.8 | 23.5 | 56.4 | 26.3 |
| Gemini-2.5-Pro | | | | | | | | | | | |
| Without Search | 10.6 | 15.6 | 11.1 | 6.0 | 12.5 | 13.9 | 0.0 | 15.8 | 5.9 | 35.1 | 0.0 |
| Image Search | 16.4 | 26.6 | 11.1 | 18.0 | 20.0 | 16.7 | 3.2 | 0.0 | 23.5 | 54.3 | 0.0 |
| Full Rollout | 23.8 | 39.1 | 14.8 | 12.0 | 27.5 | 33.3 | 6.5 | 26.3 | 29.4 | 46.8 | 13.8 |
| Full Rollout + SoM | 27.7 | 40.6 | 22.2 | 24.0 | 25.0 | 33.3 | **19.4** | 15.8 | 29.4 | 54.3 | 16.1 |
| **Open-source** | | | | | | | | | | | |
| Qwen-2.5-VL-72B | | | | | | | | | | | |
| Without Search | 0.0 | 0.0 | 0.0 | 0.0 | 0.0 | 0.0 | 0.0 | 0.0 | 0.0 | 0.0 | 0.0 |
| Image Search | 13.5 | 20.3 | 7.4 | 18.0 | 17.5 | 11.1 | 3.2 | 0.0 | 23.5 | 41.5 | 1.4 |
| Full Rollout | 6.1 | 9.5 | 7.4 | 4.0 | 5.0 | 2.8 | 3.2 | 5.3 | 11.8 | 17.0 | 1.4 |
| Full Rollout + SoM | 7.1 | 10.9 | 3.7 | 4.0 | 10.0 | 5.6 | 6.5 | 5.3 | 11.8 | 18.1 | 2.3 |

(b) **Image Search.** For each benchmark image, the top-10 image-search results are summarized by Gemini and cached. The MLLM then receives the question, the images, and these summaries as input. For multi-image questions, we append each image's summaries to the model context.

(c) **Text Search.** For text search, we retrieve the top-5 websites by default. The process is capped at 20 search rounds. In each text-search round, the model generates 3–5 refined queries

(d) **Full Rollout.** The model runs our search-agent framework and may call search tools as needed. Image-search and text-search settings match those described before. The process is capped at 20 search rounds (combining text and image search) to balance efficiency and coverage. In each image-search round, it issues exactly one query. For Qwen, pilot studies showed that more than 10 rounds yield no further gains, so we cap its search at 10 rounds.

(e) **Full Rollout + SoM.** Images overlaid with Set-of-Mark are provided to MLLMs. In addition to capabilities provided in *Full Rollout*, models can (a) call the *zoom-in* tool to examine a sub-region of an image, (b) call the *image search* tool to search with the subregion. The maximum number of tool calls is 20.

## 4 RESULTS

The results of MLLMs across search modes are presented in Table 1.

**Baseline performance.** Under *No Search* on the original MMSEARCH$^+$ benchmark, open-source **Qwen** scored **0.0%**, confirming that MMSEARCH$^+$ resisted parametric shortcuts. Closed-source models remained far from saturation (**GPT-5 10.3**, **Gemini-2.5-Pro 10.6**, **o3 15.1**). With *Image Search* (**o3 +4.2**, **GPT-5 +6.1**, **Gemini +5.8**; **Qwen +13.5**), models gained from coarse disambiguation, yet multi-hop composition remained the main blocker. With iterative image and text search,

*Full Rollout* produced large gains: **o3 36.0** (**+20.9** over no-search) and **Gemini 23.8** (**+13.2**), while **Qwen** regressed to **6.1** (**-7.4** over image-search). This indicates that the external information retrieved from search tools is essential to answering the questions correctly.

**Performance drop on easy split.**    Notably, *easy* split accuracy did *not* improve from *Image Search* to *Full Rollout*. For **o3**, we conducted a targeted error analysis on this split: there are 23 "easy" questions for which the image-only setting is correct but the full-rollout setting is not, corresponding to a 7.4-point absolute accuracy loss on our 311-task benchmark. We randomly sampled 10 of these 23 cases and manually inspected the trajectories. In 9 out of 10, the full-rollout trajectory *never* invoked image search at all: the model apparently believed it had already understood the image well enough and instead relied solely on text search or prior knowledge, thereby missing critical fine-grained visual cues. In the remaining case, the model did perform image search and retrieved relevant sources, but made a reasoning error in the final step.

These observations indicate that the drop in Easy-split accuracy is not primarily driven by over-retrieval or exposure to distractors, but by *underuse* of image search in the more complex full-rollout setting: the model sometimes "skips" a necessary image-search step when it incorrectly assumes that additional retrieval is unnecessary. We therefore attribute the Easy-split gap mainly to:

1. **Label vs. retrieval difficulty.** In MMSEARCH$^+$, "easy" meant higher likelihood of being in-model or solvable by a direct image hop, not that multi-step retrieval was trivial.

2. **Tool-use decision errors.** The dominant failure mode for **o3** is a policy-level issue—deciding *whether* to invoke image search—rather than over-retrieval or inherent drawbacks of the full-rollout setting.

Overall, rollout helped where progressive narrowing was *necessary* (academic, tech, media queries), but on easy items performance was limited by miscalibration in tool choice: the model occasionally declined to use image search even when it was crucial for success.

**Adding *SoM* (*Full Rollout + SoM*) gave consistent, targeted gains.** The improvements were **o3 37.6** (**+1.6**), **Gemini 27.7** (**+3.9**), **Qwen 7.1** (**+1.0**) over *Full Rollout*. Set-of-Mark anchored reasoning on fine-grained visual cues (micro-text, logos, key entities) before expanding the search space.

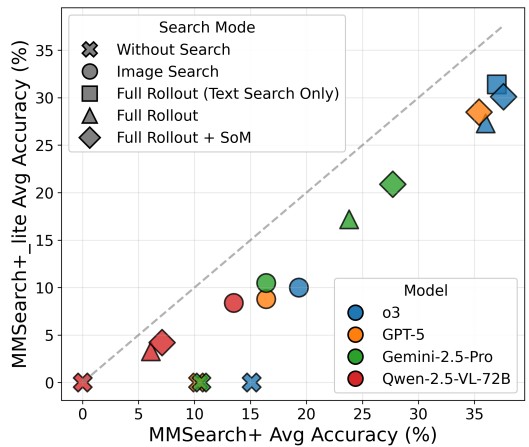

Figure 4: The differences of method performance on MMSEARCH$^+$ and MMSEARCH$^+_{lite}$. The closer a point is to the diagonal line $y = x$, the smaller the influence of internal knowledge.

**Decoupling internal knowledge.**    To disentangle the effect of models' internal knowledge from their use of external tools, we further evaluate on MMSEARCH$^+_{lite}$, a subset of 239 tasks that are unsolvable without search by any model we tested (Table 4). The relationship between performance on the full and lite benchmarks is visualized in Figure 4.

On MMSEARCH$^+_{lite}$, the best-performing setting is full rollout with only text search (abbreviated as *Text Search* in the table) with the **o3** model, achieving 31.4% accuracy. This suggests that, once questions solvable purely from internal memory are removed, current MLLMs still struggle to exploit fine-grained image search effectively. In many categories, we observe a drop in accuracy when moving from only text search to full rollout / full rollout + SoM. However, there remain categories where image search is clearly beneficial: for **o3** in *Sports*, accuracy increases by 6.6 (full rollout) and 10.0 points (full rollout + SoM) over text-only rollout. Overall, most methods lie close to the diagonal in Figure 4, indicating that the qualitative trends observed on MMSEARCH$^+$ are not driven primarily by parametric memorization, but by genuine tool-use and retrieval behavior.

# 5 ANALYSIS

We present a list of observations (Section 5.1) and an error analysis (Section 5.2).

## 5.1 OBSERVATIONS

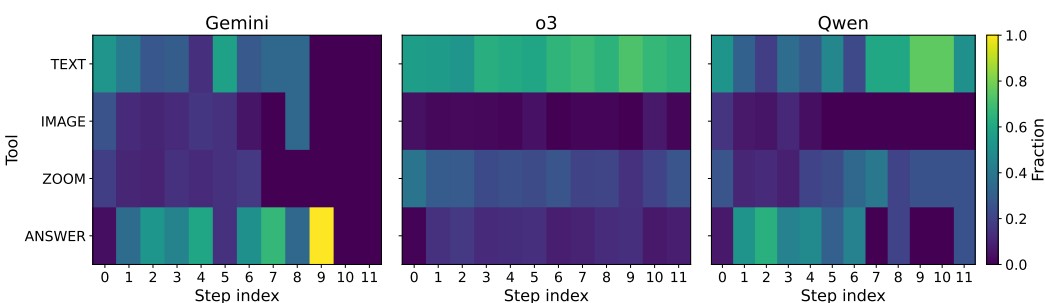

Figure 5: Stepwise tool distribution for Gemini, o3 and Qwen under *Full Rollout+SoM*, capped at Step 12 for visualization. We use search round and step interchangeably, and an MLLM can make multiple search queries in one round (details in ξ 3).

**Closed-source models retained broad geographic knowledge.** They often recognized rare locales and supplied plausible context; geography tasks were comparatively strong. Without search tools, o3 achieved 31.2%, while human annotators with a standard browser only achieved 20.3%. On other categories like sports and Vlog, the human baseline usually outperformed the tested MLLMs. We name this phenomenon the different comparative advantages of humans and MLLMs.

**o3 exhibited strong long-context coordination, while Gemini leveraged fewer search calls.** On the one hand, o3 sustained 10+ search rounds and reasoned over $\sim 50+$ retrieved items without excessive redundancy, contributing to its rollout gains. On the other hand, Figure 5 (left) shows that Gemini always answered on or before Step 9. However, more tool calls did not correlate with trajectory correctness: Figure 15 shows that incorrect trajectories typically contained more search calls than correct ones.

**Format correctness of closed-source models' tool calls, and Qwen's tool-use instability.** All the closed-source models consistently produced valid tool calls. For Qwen, we observed 421 invalid image-search calls across 45 of 311 tasks, inflating retries without quality improvements.

**More text search calls than image search calls, and zoom-in operations were not always followed by a sub-image search call.** In Figure 5, all models made more text-search calls than image-search calls. We analyze the contribution of the "zoom-in then image-search" tool-call pattern to the final accuracy. For Gemini-2.5-Pro, out of 39 newly answered correctly questions from full rollout to full rollout + SoM, there are 5 such cases in which the model zoomed in on an image and then performed an image search (12.8%). For o3, out of 42 newly answered correctly questions, there is 1 such case (2.4%). This indicates that Gemini utilized fine-grained image search more frequently, while o3 often zoomed in simply to view a region more clearly.

To empirically support this observation, we examine the Markov transition probabilities between tool types (Figure 12). In particular, Set-of-Mark allows cropping and sub-image search, so we measure how often a zoom action is immediately followed by an image search. The transition probability $P(\textbf{image} \mid \text{zoom})$ varies substantially across models: Gemini uses image search after zooming in 25.37% of cases (34/134), Qwen in 10.56% (15/142), and o3 in only 2.87% (18/627). These statistics are consistent with the accuracy gains above—Gemini tends to use zoomed-in regions as retrieval queries, whereas o3 seldom does, often preferring further zooms or text search instead.

**Image search and text search benefit different task categories.** The accuracies of o3 with text search (up to 10 rounds) and with full rollout are similar (37.0 vs. 36.0). However, their category-

level accuracy distributions differ substantially. For example, image search yields better performance in *Film/TV*, whereas additional text search rounds provide greater benefits in *Geography*.

**Scene-text is predominantly processed as textual evidence rather than visual evidence.** Across tasks containing logos, signage, or other forms of scene text, we observed that MLLMs overwhelmingly chose to convert visual text into explicit textual queries rather than invoking image search on the corresponding subregions. By detecting scene text with DeepSeek-OCR and manually inspecting sampled trajectories, we found that both o3 and Gemini-2.5-Pro typically relied on their built-in OCR capabilities to extract strings and issue text-search calls. For o3, 8 out of 10 sampled trajectories used only text search, while Gemini behaved similarly in 7 of 10 cases. Direct image-search calls on SoM-highlighted text regions were rare and generally appeared only when the model seemed uncertain about the exact transcription or wished to leverage additional visual cues such as layout or font. These findings suggest that, within MMSearch-Plus, scene-text reasoning is mediated primarily through textual grounding rather than visual retrieval, highlighting that multimodal agents implicitly choose between two tool pathways, (a) OCR + text search versus (b) image search, based on confidence and perceived utility.

## 5.2 ERROR ANALYSIS

We categorize failure cases using a manually curated taxonomy and report the distribution over categories in Figure 6. Labels are assigned by the authors (single dominant label per erroneous prediction) for **Gemini-2.5-Pro** under the *full-rollout* setting. Examples of incorrect task trajectories are provided in Figure 7.

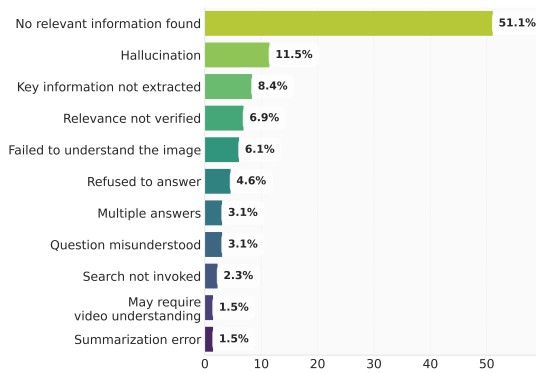

Figure 6: **Distribution of error types.** Annotated on Gemini-2.5-Pro (full rollout).

**High-frequency categories.** The two most prevalent failure modes are *no relevant information found* and *hallucination*, responsible for $51.1\%$ and $11.5\%$ of errors, respectively ($62.6\%$ combined). The former typically reflects a retrieval failure: noisy or overly specific queries miss on-topic pages. The latter reflects vision-driven misattribution: the model latches onto an incorrect event from internal knowledge and either does not invoke search or discounts contradictory evidence in the retrieved pages.

**Other observed categories.** We also observe failures from misinterpreting the multimodal context; hallucinations grounded in either images or parametric knowledge; refusals or multi-answer outputs when evidence is insufficient to disambiguate a single event; failing to invoke search (falling back on outdated priors); and errors when summarizing relevant webpages.

**Qualitative takeaways and implications.** Two recurring patterns emerge: (i) there is no single "golden path"—agents can succeed via varied orders of search/crop/verify; and (ii) stronger multimodal internal knowledge narrows the practical search space, enabling targeted verification over broad exploration. While augmenting multimodal internal knowledge can reduce exploration overhead, end-to-end reliability still depends on robust verification and tighter cross-modal grounding.

## 6 RELATED WORK

Classical vision–language benchmarks probe compositional and commonsense reasoning in images and text (Johnson et al., 2017; Krishna et al., 2016; Zellers et al., 2019; Mathew et al., 2021). Among these, GQA (Hudson & Manning, 2019) emphasizes structured compositional reasoning using real-world scenes, while OK-VQA (Marino et al., 2019) requires external knowledge beyond the visual signal. More recently, InfoSeek (Chen, 2023) frames VQA as an information-seeking task, but relies on offline retrieval corpora whose contents frequently overlap with pretraining data for modern

MLLMs, weakening its ability to measure genuine retrieval behavior. LiveVQA (Fu et al., 2025) benchmarks *live* visual knowledge by deriving VQA instances from recent web content beyond models' cutoff dates, and shows that multimodal search augmentation substantially boosts performance.

Beyond static images, recent benchmark suites extend to long videos and interleaved modalities (Wu et al., 2024; Li et al., 2024; Hu et al., 2025; Jeong et al., 2025). Concurrent work on web agents studies planning, browsing, and evidence gathering in open worlds (Krishna et al., 2024; Wei et al., 2025; Geng et al., 2025; Li et al., 2025a; Tao et al., 2025).

Multimodal web search benchmarks begin to couple vision with browsing but often emphasize entity matching or template-based grounding over fine-grained visual reasoning (Jiang et al., 2024; Wu et al., 2025). Closest to our setting, MM-BrowseComp targets image/video evidence encountered during search and reveals substantial headroom even for frontier models (Li et al., 2025b).

A complementary line promotes "thinking with images" via improved grounding or RL-style training to elicit image-first reasoning (Wu & Xie, 2024; Zheng et al., 2024; Fan et al., 2025; Zheng et al., 2025b; Su et al., 2025a; Xia et al., 2025).

We differ from prior benchmarks by targeting *sparse*, spatial–temporal visual cues that must be transformed into targeted queries and multi-step browsing plans. Crucially, tasks in MMSearch-Plus are designed to be extremely unlikely to be answerable from internal model knowledge alone, requiring models to (i) decide what visual

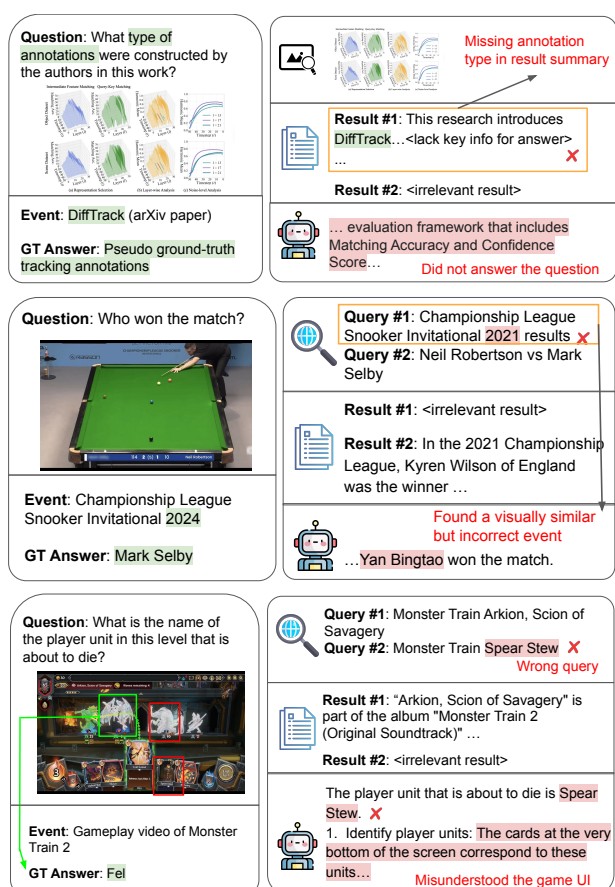

Figure 7: **Representative error case studies.** Each row illustrates an example from a high-frequency error type (Figure 6). Panels show the prompt, GT answer, and a shortened rollout. From top to bottom: (a) *Key info not extracted.* (b) *Relevance not verified.* (c) *Question misunderstood.*

regions matter, (ii) choose appropriate external tools (image vs. text search), and (iii) gather and verify evidence online. This provides a focused stress test for agentic multimodal browsing systems.

## 7 CONCLUSION

This work introduces **MMSearch-Plus**, a benchmark that restores the genuinely multimodal pressures of web browsing by requiring agents to extract weak, localized visual cues and propagate them through noisy retrieval to out-of-image facts. Our **Spatial–Temporal Extrapolation** curation procedure preserves the persistence and difficulty profile of text-only browsing while compelling fine-grained visual reasoning and provenance verification. Evaluating a range of open and closed MLLMs within a model-agnostic search agent framework reveals a substantial headroom: the strongest agent we tested (o3) achieved less than 40% accuracy, while a strong open-source model (Qwen-2.5-VL-72B-Instruct) achieved 13.5%. These results indicate that current systems struggle to (i) decide when and how to crop and reuse visual evidence, and (ii) maintain verifiable chains of evidence across long-horizon tool use. We hope MMSearch-Plus serves as a rigorous stress test and a common yardstick for progress on multimodal browsing.

ETHICS STATEMENT

**Principles.** We follow the Code of Ethics: uphold scientific rigor, avoid harm, ensure transparency, foster fairness, respect intellectual labor, and protect privacy and confidentiality.

**Data sources, consent, and licensing.** We curate only publicly accessible material from (i) consumer video platforms (keyword queries) and (ii) open-access repositories (e.g., arXiv). We comply with platform Terms of Service and licenses and preserve attributions (e.g., video URL, paper URL). Items are treated as "events" and used solely for research.

**Privacy, minimization, and redaction.** We mask sensitive content (e.g., faces, license plates); we exclude minors and content revealing private spaces when feasible. Residual risks remain; we provide an appeals/takedown channel.

**Fairness.** To counter skews in public platforms, we conducted manual sampling across predefined categories and regions, report measurable distributions, and forbid uses that target protected classes or amplify inequities.

**Limitations.** While diverse, our data inherits biases from public web sources and the domains we sampled; some tasks still under-represent non-English or low-visual-density pages. Our evaluation focuses on general-purpose agents with standard tool access; specialized systems with richer UI control or domain knowledge may behave differently. Finally, we primarily target image–text browsing; videos and dynamic interfaces remain underexplored.

REPRODUCIBILITY STATEMENT

We have taken several steps to facilitate independent verification of our results. Implementation details, hyperparameters, and evaluation protocols are documented in Section 3. We uploaded a randomly selected subset (seed = 42, 30 task examples) as supplementary material due to the 100MB space constraint, and released an anonymized archive on how to use the dataset in `https://anonymous.4open.science/r/MMSearch-Plus-D54E`.

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

## A  USE OF LLMs

The manuscript text was polished for fluency using Grammarly after the authors completed a full draft. The authors then manually reviewed all suggestions to ensure their appropriateness.

## B  EXAMPLE FROM MMSEARCH-PLUS

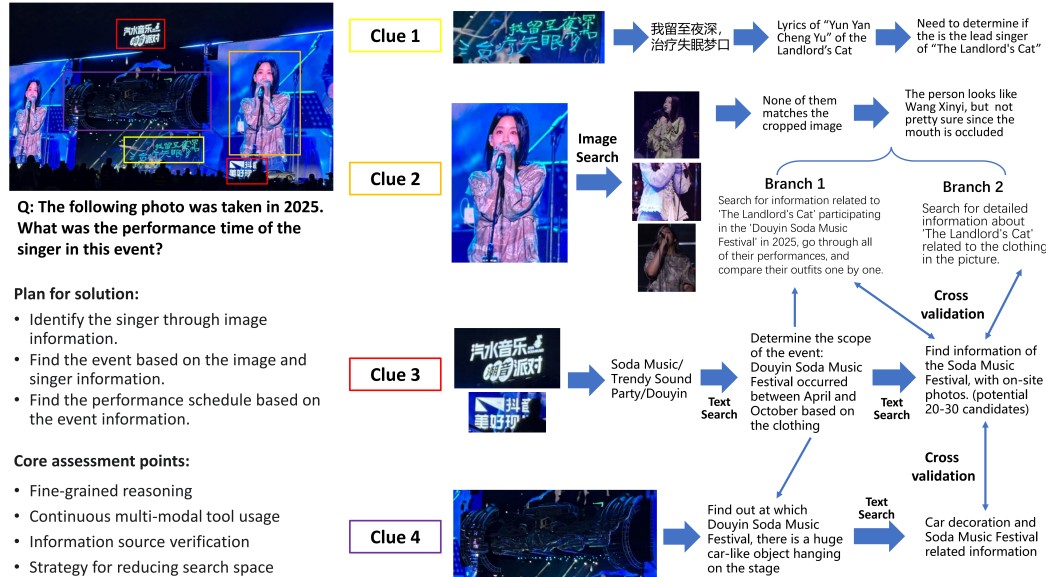

Figure 8: **Example MMSearch-Plus item.** Given a 2025 concert photo and the query "What was the singer's performance time?", the agent's golden path extracts localized cues—micro-text/lyrics, a crop of the singer, festival/brand signage, and a distinctive stage prop—and issues targeted image/text searches that (i) identify the artist/outfit, (ii) resolve the event (e.g., a specific festival and city/date), and (iii) retrieve and *cross-validate* the official schedule to obtain the exact performance time. The answer is not explicitly written in the prompt or the image, underscoring MMSearch-Plus's emphasis on fine-grained multimodal reasoning with provenance checks under retrieval noise.

## C  COMPARISON WITH PREVIOUS BENCHMARKS

Table 2: Comparison of multimodal information–seeking datasets and benchmarks, ordered by release date.

| Benchmark/ Dataset | MM | Search | | | Multi-hop (>2) | Thinking w/ Image | Multi-Image |
|---|---|---|---|---|---|---|---|
| | | Text | Image | Multi-turn | | | |
| InfoSeek (Chen, 2023) | ✓ | ✓ | ✓ | ✗ | ✗ | ✗ | ✗ |
| FRAMES (Krishna et al., 2024) | ✗ | ✓ | ✗ | ✓ | ✓ | ✗ | ✗ |
| MMSearch (Jiang et al., 2024) | ✓ | ✓ | ✓ | ✗ | ✗ | ✗ | ✗ |
| FactualVQA(Wu et al., 2025) | ✓ | ✗ | ✓ | ✗ | ✗ | ✗ | ✗ |
| ChineseSimpleVQA (Gu et al., 2025) | ✓ | ✗ | ✗ | ✗ | ✗ | ✗ | ✗ |
| BrowseComp (Wei et al., 2025) | ✗ | ✓ | ✗ | ✓ | ✓ | ✗ | ✗ |
| BrowseComp-ZH (Zhou et al., 2025) | ✗ | ✓ | ✗ | ✓ | ✓ | ✗ | ✗ |
| BrowseComp-VL (Geng et al., 2025) | ✓ | ✓ | ✓ | ✓ | ✓ | ✗ | ✗ |
| MM-BrowseComp (Li et al., 2025b) | ✓ | ✓ | ✓ | ✓ | ✓ | ✓ | ✗ |
| **Ours** | ✓ | ✓ | ✓ | ✓ | ✓ | ✓ | ✓ |

# D  RELATED WORK

We present a more comprehensive literature review in the Appendix.

**Multimodal Reasoning Benchmarks.** Classical vision-and-language benchmarks such as CLEVR (Johnson et al., 2017), VCR (Zellers et al., 2019), Visual Genome (Krishna et al., 2016), and InfographicVQA (Mathew et al., 2021) established core evaluations for compositional reasoning, commonsense inference, and visual question answering. Recent efforts extend these settings to longer contexts and interleaved modalities, including LongVideoBench (Wu et al., 2024), MVBench (Li et al., 2024), Video-MMMU (Hu et al., 2025), and VideoRAG (Jeong et al., 2025), which highlight long-context understanding and retrieval augmentation. Other work such as MIRB (Zhao et al., 2024), pixel-level VCR (Zhang et al., 2024), and ReasonMap (Feng et al., 2025) aim to connect perception with explicit reasoning. However, these datasets are primarily self-contained and do not require dynamic interaction with the open web or tool-mediated search.

**Web Browsing and Search-Enhanced Agents.** A large body of work explores language models as web agents that plan, browse, and gather evidence. Earilier works like FRAMES (Krishna et al., 2024) focus on multi-hop QAs on Wikipedia-like data. More recently, benchmarks such as BrowseComp (Wei et al., 2025) and its Chinese variant BrowseComp-ZH (Zhou et al., 2025) test multi-step persistence in text-heavy web navigation, where even frontier models struggle. Open-source agents including WebSailor (Li et al., 2025a), WebShaper (Tao et al., 2025), and WebWatcher (BrowseComp-VL) (Geng et al., 2025) emphasize robustness, data synthesis, and limited multimodal inputs.

In parallel, reinforcement-learning–based frameworks push the boundaries of search adaptivity and depth. Search-R1 (Jin et al., 2025), R1-Searcher (Song et al., 2025), ReSearch (Chen et al., 2025), Pangu DeepDiver (Shi et al., 2025), and DeepResearcher (Zheng et al., 2025a) train persistence and decomposition skills through reward shaping. ASearcher (Gao et al., 2025) extends this direction with large-scale asynchronous reinforcement learning, decoupling exploration from evaluation to unlock long-horizon search over dozens of hops. Complementary agentic systems such as Open Deep Search (Alzubi et al., 2025), AutoAgent (Tang et al., 2025), and Search-o1 (Li et al., 2025c) further democratize and automate search-enhanced reasoning.

**Multimodal Web Search and MM-BrowseComp.** Beyond text-only browsing, MM-Search (Jiang et al., 2024) and MMSearch-R1 (Wu et al., 2025) introduce image-grounded queries, but many tasks still reduce multimodality to entity matching rather than genuine visual reasoning. MM-BrowseComp (Li et al., 2025b) addresses this gap with a 224-question benchmark designed for multimodal browsing agents where key evidence may be embedded in images or videos encountered during search. The benchmark provides a verified checklist for fine-grained analysis of multimodal dependencies and reasoning paths, and reveals that even top models (e.g., OpenAI o3 with tools)

achieve only $\sim 29\%$ accuracy, underscoring substantial headroom for native multimodal reasoning within browsing pipelines. Our setting is closely related to, and concurrent with, MM-BrowseComp, but differs in focusing on *sparse* visual cues that must be transformed into targeted queries rather than predominantly entity-level matching.

**Thinking with Images.** A growing body of work argues that multimodality should go beyond tool orchestration to *think with images*. Methods such as V\* (Wu & Xie, 2024) and instruction-guided masking (Zheng et al., 2024) improve visual grounding inside LLMs. RL-style training further enforces image-first reasoning: GRIT (Fan et al., 2025), DeepEyes (Zheng et al., 2025b), Pixel Reasoner (Su et al., 2025a), and Visionary-R1 (Xia et al., 2025) encourage explicit visual deliberation before action. Visual Agentic Reinforcement Fine-Tuning (Liu et al., 2025), OpenThinkIMG (Su et al., 2025b), and One-RL-to-See-Them-All (Ma et al., 2025) propose unified training frameworks for multimodal agents, while broader systems like PyVision (Zhao et al., 2025) and Thyme (Zhang et al., 2025) generalize the paradigm.

**Positioning and Summary.** Taken together, web-browsing benchmarks stress persistence and decomposition in open-world settings (Wei et al., 2025; Li et al., 2025a; Tao et al., 2025; Geng et al., 2025), while multimodal benchmarks emphasize perception-heavy reasoning (Johnson et al., 2017; Krishna et al., 2016; Zellers et al., 2019; Mathew et al., 2021). Recent multimodal browsing datasets—including MM-BrowseComp (Li et al., 2025b)—begin to unite these threads but still leave open how *sparse* visual evidence should drive query formulation and multi-step planning. We formulate the Spatial-Temporal Extrapolation methodology to concretely design challenging VQAs for testing multimodal search abilities. We also stress the importance of grounding and cropping images to allow fine-grained multimodal reasoning.

## E  DETAILED DATASET STATISTICS

The distributions of primary categories across MMSearch-Plus are reported in Table 3.

## F  SET-OF-MARK IMPLEMENTATION DETAILS

Figure 9 shows a list of task images overlaid with Set-of-Mark bounding box annotations from MMSearch-Plus.

Table 3: Primary category distribution.

| Category | Count | % |
|---|---|---|
| Geography | 64 | 20.6% |
| Sports | 54 | 17.4% |
| Academic Research | 50 | 16.1% |
| Film & TV | 40 | 12.9% |
| Technology | 36 | 11.6% |
| Video Games | 31 | 10.0% |
| Vlog | 19 | 6.1% |
| Music | 17 | 5.5% |

In our context, set-of-mark for a task image consists of bounding boxes of key visual entities. Each box has a unique index. In the first user turn, we present both the raw image and an overlaid version with box outlines and indices, avoiding occlusion while allowing cross-reference. The framework exposes:

- `zoom_in(index)`: return a cropped sub-image for localized reasoning;
- `image_search(index)`: launch an image search seeded by the cropped region (compatible with the standard image-search pathway).

We include SoM because baseline MLLMs were unreliable at proposing visually precise crops *from scratch*. Providing accurate marks reduces the burden on detection/localization and lets models focus on reading, matching, and reasoning. While automatic mark generation (e.g., strong open-vocabulary detectors) is conceivable, robust coverage is challenging due to long-tail entities common in our benchmark; therefore MMSearch-Plus uses human-curated boxes.

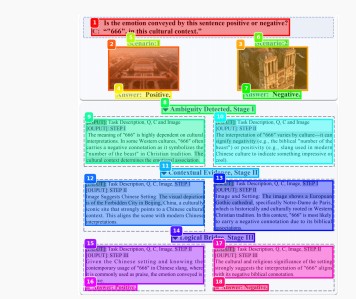

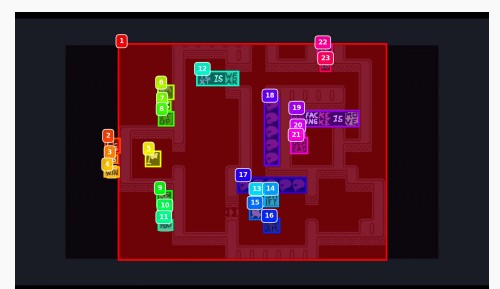

**Q:** What is the title of this paper?
**A:** MUCAR: Benchmarking Multilingual Cross-Modal Ambiguity Resolution for Multimodal Large Language Models

**Q:** What is the name of the previous level?
**A:** Exercise Hall

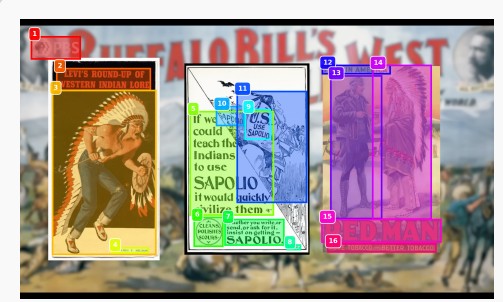

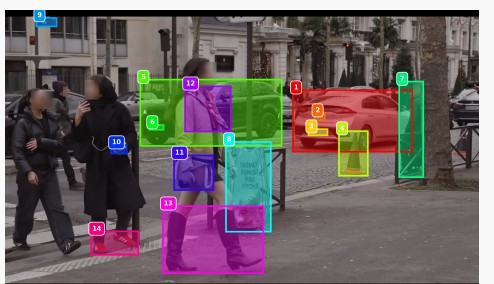

**Q:** What is the title of this video?
**A:** Who Can Identify as a Native American?

**Q:** In Paris Fashion Week, what brand did the person in the image represent? In which year did the event occur, and what is the event's name?
**A:** StreetStyle Winter 2025 Hermes

Figure 9: Examples of task images overlaid with Set-of-Mark bounding box annotations from MMSearch-Plus.

# G    EVALUATION DETAILS

We initially tested both GPT-4o and Gemini-2.5-Pro as judges and found their outputs consistent. The subsequent experiments used GPT-4o. To further support reproducibility, we optionally provide lightweight rule-based verification methods.

# H  RESULTS ON MMSEARCH$^+_{lite}$

To eliminate the impact of model's internal knowledge, we obtain MMSEARCH$^+_{lite}$ from MMSEARCH$^+$ by removing all tasks from MMSEARCH$^+$ answerable without search by any model we tested. The results are reported in Table 4 and visualized in Figure 10 and 4.

In MMSEARCH$^+_{lite}$, the best performing setting is full rollout with only text search (abbreviated as *Text Search* in the table) with o3 model, achieving 31.4% accuracy. This suggests that the current MLLMs cannot cleverly utilize fine-grained image search. In many categories, we observed a drop in accuracy from only text search to full rollout/full rollout + SoM. However, in certain categories like *Sports*, there is a great benefit of image search: for o3, the accuracy increases by 6.6 (full rollout) and 10.0 points (full rollout + SoM).

Table 4: End-to-end results on the MMSEARCH$^+_{lite}$ benchmark. All numbers are accuracy (%). Darker blue means higher accuracy. The best and second-best methods under each setting are shown in bold and underline, respectively.

| Model / Search Mode | Avg | By Category | | | | | | | |
|---|---|---|---|---|---|---|---|---|---|
| | | Geo. | Sports | Acad. | Film/TV | Tech | Games | Vlog | Music |
| **Human** | | | | | | | | | |
| Browser | 18.8 | 11.6 | 26.7 | 17.9 | 33.3 | 22.0 | 20.7 | 6.7 | 26.7 |
| **Closed-source** | | | | | | | | | |
| o3 (2025-04-16) | | | | | | | | | |
| Without Search | 0.0 | 0.0 | 0.0 | 0.0 | 0.0 | 0.0 | 0.0 | 0.0 | 0.0 |
| Image Search | 10.0 | 14.0 | 13.3 | 10.3 | 16.7 | 4.9 | 17.2 | 3.3 | 0.0 |
| Text Search | **31.4** | 44.2 | 16.7 | **41.0** | **25.0** | **29.3** | **37.9** | 16.7 | 26.7 |
| Full Rollout | 27.3 | **46.5** | 33.3 | 17.9 | 8.3 | 14.6 | **37.9** | 16.7 | **33.3** |
| Full Rollout + SoM | 30.1 | 44.2 | **36.7** | 30.8 | 16.7 | 24.4 | 34.5 | 16.7 | 20.0 |
| GPT-5 | | | | | | | | | |
| Without Search | 0.0 | 0.0 | 0.0 | 0.0 | 0.0 | 0.0 | 0.0 | 0.0 | 0.0 |
| Image Search | 8.8 | 9.3 | 13.3 | 7.7 | **25.0** | 4.9 | 13.8 | 3.3 | 0.0 |
| Full Rollout + SoM | 28.5 | **46.5** | **36.7** | 23.1 | 8.3 | 24.4 | 34.5 | 6.7 | **33.3** |
| Gemini-2.5-Pro | | | | | | | | | |
| Without Search | 0.0 | 0.0 | 0.0 | 0.0 | 0.0 | 0.0 | 0.0 | 0.0 | 0.0 |
| Image Search | 10.5 | 14.0 | 13.3 | 12.8 | **25.0** | 7.3 | 10.3 | 3.3 | 0.0 |
| Full Rollout | 17.2 | 4.7 | 23.3 | 28.2 | **25.0** | 12.2 | 27.6 | 6.7 | 20.0 |
| Full Rollout + SoM | 20.9 | 23.3 | 13.3 | 28.2 | **25.0** | 19.5 | 24.1 | **20.0** | 6.7 |
| **Open-source** | | | | | | | | | |
| Qwen-2.5-VL-72B | | | | | | | | | |
| Without Search | 0.0 | 0.0 | 0.0 | 0.0 | 0.0 | 0.0 | 0.0 | 0.0 | 0.0 |
| Image Search | 8.4 | 14.0 | 10.0 | 7.7 | **25.0** | 2.4 | 10.3 | 3.3 | 0.0 |
| Full Rollout | 3.3 | 4.7 | 3.3 | 0.0 | 8.3 | 4.9 | 3.4 | 3.3 | 0.0 |
| Full Rollout + SoM | 4.2 | 2.3 | 6.7 | 2.6 | 8.3 | 4.9 | 3.4 | 6.7 | 0.0 |

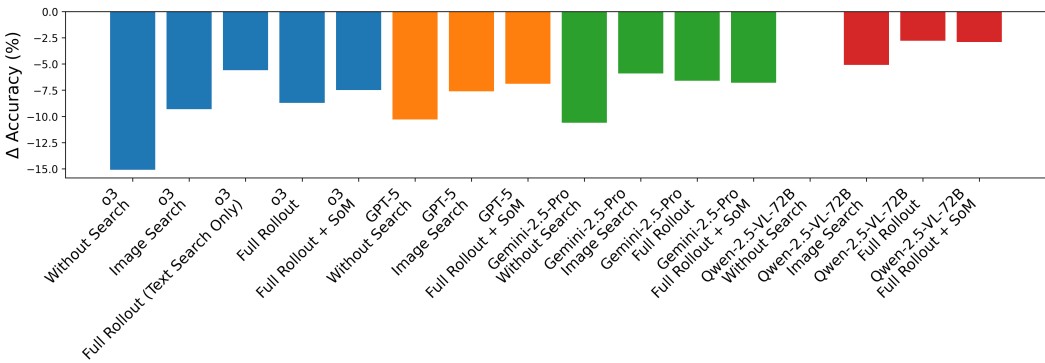

Figure 10: The differences of method performance on MMSEARCH$^+$ and MMSEARCH$^+_{lite}$. More negative means a larger influence of model's internal knowledge.

## I AGENT BEHAVIOR ANALYSIS

This appendix presents detailed visualizations of tool usage patterns across the three multimodal agents: Gemini, o3, and Qwen. All figures show episodes capped at 20 steps for readability.

### I.1 EPISODE RASTER PLOTS

Figure 11 shows the tool usage patterns for the 40 longest episodes from each agent. Each row represents one episode, sorted by length (longest at top). Columns represent sequential steps, with colors indicating tool types: text search, image search, zoom-in, and answer. Dark blue strips represent an MLLM did not fully utilize 20 rounds and stopped early.

### I.2 TOOL TRANSITION MATRICES

Figure 12 shows the Markov transition probabilities between tool types.

In particular, Set-of-Mark allows image cropping and sub-image searching. Therefore, we investigate whether many of the cropped images were searched in the immediate next round, i.e., whether $P(\mathbf{image}|zoom)$ is large.

- Gemini: 25.37% (34 out of 134 ZOOM actions)
- Qwen: 10.56% (15 out of 142 ZOOM actions)
- o3: 2.87% (18 out of 627 ZOOM actions)

### I.3 ZOOM-IN BEHAVIOR ANALYSIS

Figures 13 and 14 analyze zoom-in behavior patterns across agents. o3 shows the most aggressive zooming behavior, while Qwen and Gemini are more conservative.

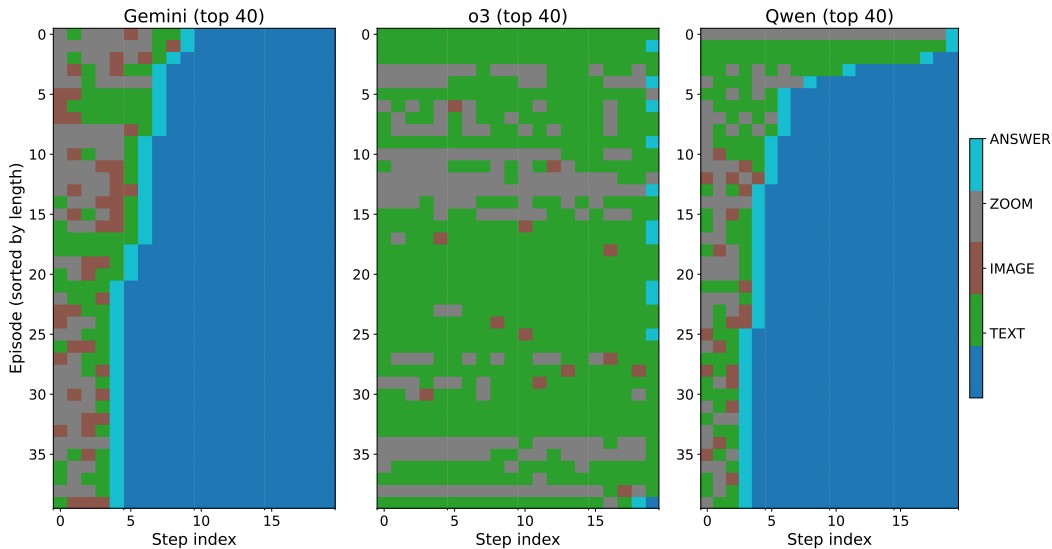

Figure 11: Episode raster plots showing tool usage patterns for the 40 longest episodes per agent. Each row is an episode (sorted by length), each column is a step.

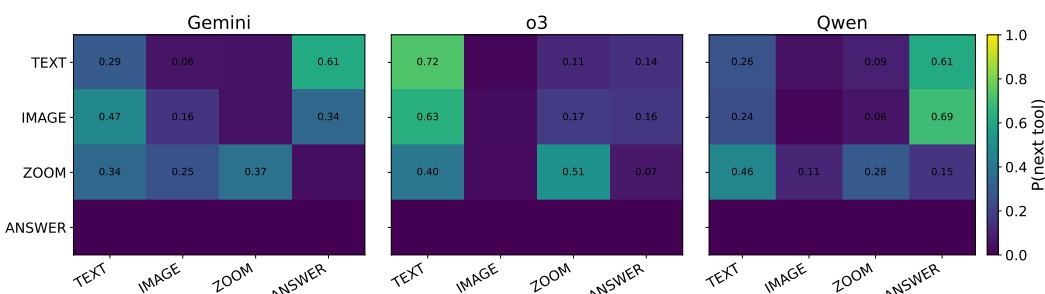

Figure 12: Tool-call transition matrices showing probabilities $P(\text{next tool}|\text{current tool})$. Rows and columns represent current tools and next tools, respectively. Values $\geq 0.05$ are annotated. Note that ANSWER is terminal (bottom row is zeros).

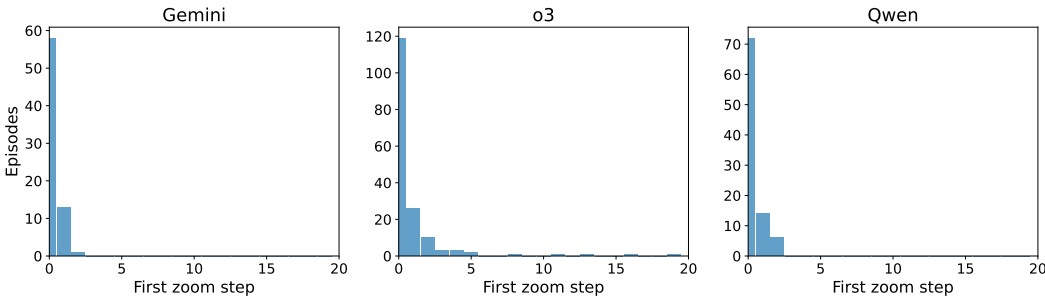

Figure 13: Distribution of first zoom-in step positions. o3 tends to zoom earlier in episodes compared to Gemini and Qwen.

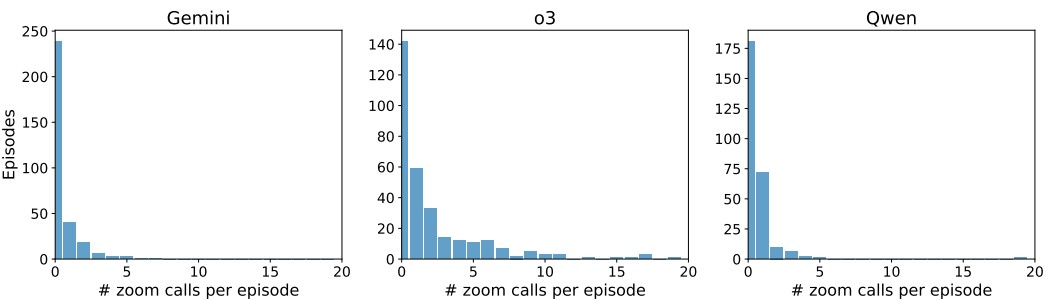

Figure 14: Distribution of zoom-in frequency per episode. o3 exhibits the highest zoom-in frequency, with some episodes containing 10+ zoom operations.

## J   ROLLOUT TRAJECTORY STATISTICS

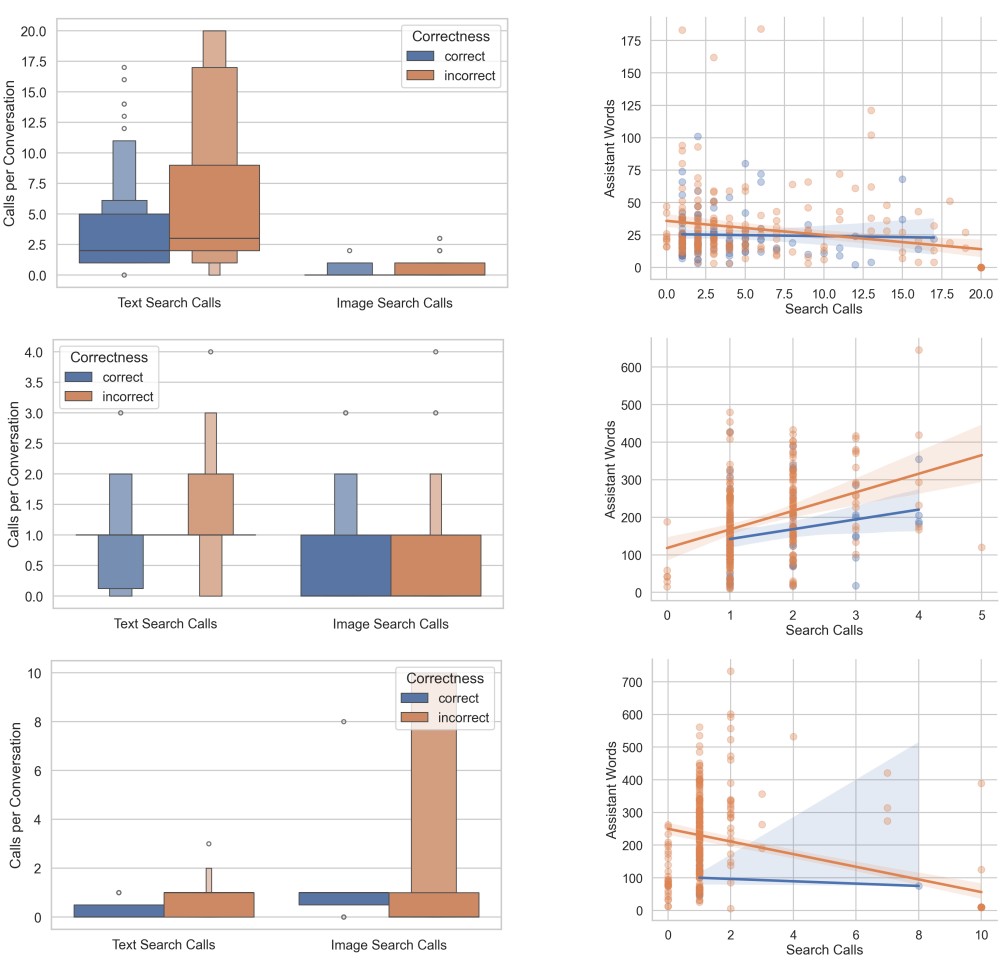

Figure 15: Reasoning-trajectory statistics. Rows: MLLMs (o3, Gemini, Qwen from top to down). Columns: left—distribution of image search calls and text search calls per trajectory; right—number of words in an MLLM's response versus the number of search calls, stratified by correctness. Statistics collected in *full rollout* mode.

Figure 15 shows the distribution of number of search calls for correct and incorrect task trajectories (left) and the relation of the number of words in a model's trajectory (excluding the words for tool calls) vs. the number of search calls (right) under the *Full Rollout* setting.

## K   GOOD CASES

**Task:**

Excluding the games released through the company's white-label publishing service, what games did the publisher of the game shown in the image release in 2024?

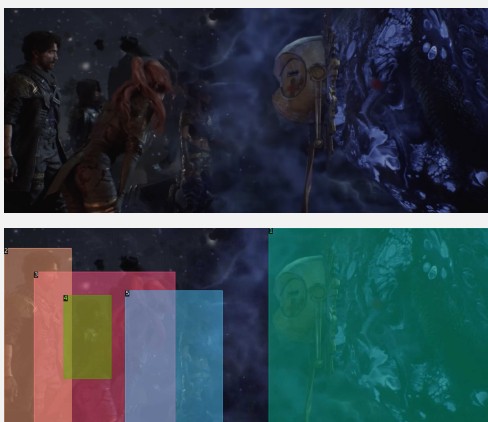

**assistant:**

**Tool Call:** `image_zoom_in_tool`

```
{
  "bbox_index": 2,
  "image_name": "n5x8zt8dok.png"
}
```

**tool:**

The cropped sub-image is provided in the next user round.

**user:**

`<zoom_in_image_name>` n5x8zt8dok_round_1_crop_1.png `</zoom_in_image_name>`
`<image_zoom_in_result>`

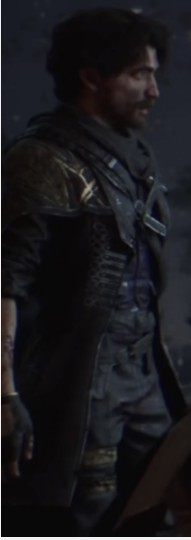

`</image_zoom_in_result>`

**assistant:**

**Tool Call:** `image_search`

```
{
  "image_name": "n5x8zt8dok_round_1_crop_1.png"
}
```

**tool:**

```
<image_search_result>
```
## Result 1
**web_info:** Based on the provided web content, it is not possible to answer your question. The text is from the Douyin (the Chinese version of TikTok) website and contains legal information, site navigation links, user agreement details, and contact information for the platform itself. There is no mention of a specific game, a game publisher, or a list of games released in 2024.

**related_info:** The reference image and the search result demonstrate a very high degree of relevance. Visually, the central character in both images is identical, sharing the same facial features, beard, and distinctive dark coat, indicating they are the same individual from the same source material. The reference image offers a more subdued profile view, while the thumbnail presents the character in a dynamic action scene, which strongly suggests a shared origin, likely a video game or animation. This visual consistency is further cemented by the semantic content of the search result's title. The title explicitly names a work, Light and Shadow: Expedition 33, and the text overlay on the thumbnail perfectly matches the first part of the title, creating a seamless link between the visual and textual information. The search result effectively takes the ambiguous character from the reference image and places them within a specific, named context. Therefore, the search result is exceptionally relevant as it not only contains the exact same core visual element but also enriches it with crucial identifying information about the production it belongs to.

## Result 2
**web_info:** The provided web page does not contain the information required to answer your question. The page announces the game *Clair Obscur: Expedition 33*, identifying it as a turn-based RPG set for release in 2025 on console and PC, and will be available on XGP from its launch day. However, the text does not name the game's publisher. Therefore, it does not provide a list of other games released by this publisher in 2024, nor does it mention anything about a white-label publishing service.

**related_info:** The visual features show an extremely high degree of matching, as the male character in the reference image is identical to the one in the thumbnail, from his hairstyle and beard to the specific design of his dark coat with gold accents on the shoulders. The reference image appears to be a close-up or a cropped frame, while the thumbnail provides a wider shot of the same character in a desolate, ruin-like environment, suggesting they originate from the same source material. The title, "Clair Obscur: Expedition 33," semantically anchors these visuals, explicitly identifying them as content from a newly announced video game. This creates perfect consistency between the visual elements and the textual information. Therefore, the search result is exceptionally relevant to the reference content. The two are not just related in topic but are direct components of the same subject, with the search result providing the precise name and broader context for the character shown in the reference image, making it a definitive and highly informative match.

## Result 3
**web_info:** (Chinese results abbreviated)
**related_info:** The analysis reveals an extremely high degree of relevance between the reference content and the search result. Visually, the core subject, a male video game character, is a perfect match. The thumbnail features the exact same character as the reference image, with identical dark clothing, hairstyle, and overall design, simply captured in a wider shot that includes another character and subtitles. This strong visual consistency is further solidified by the semantic information in the title. The title, "Gustave Argument Scene Breakdown...", explicitly names the character as "Gustave," which aligns with the subtitles in the thumbnail and provides crucial context that the reference image lacks. The topic overlap is absolute, as both the reference and the result are centered on this specific character from what is clearly the same video game. The search result does not just show the same character but offers a deeper dive into a specific scene he is in. Consequently, the search result is exceptionally relevant, directly expanding upon the subject of the reference image by providing specific narrative context and analysis.

## Result 4

**web_info:** Based on the provided webpage content, there is no information about games released by the publisher 'gamescardnet' in 2024. The content is from an Instagram page for 'gamescardnet' that features posts about various video games.

The primary post, from March 18th, is about the future release "Wwe 2K25". Other posts shown on the page also feature upcoming or undated games such as "NBA 2K26", "Cronos The New Dawn", "EA Sports NHL 26", "EA Sports FC 25", "Hell is Us", and "Mafia The Old Country". The page also includes a post for "Spider-Man 2", which was released prior to 2024.

The provided text does not contain a list of games released in 2024. Furthermore, there is no information regarding a "white-label publishing service".

**related_info:** The analysis reveals a very strong visual match between the reference content and the thumbnail, as both images clearly depict the same male video game character, identifiable by his facial features, hairstyle, and distinctive fantasy-style clothing. The thumbnail is simply a higher-quality, more dramatic shot of the individual shown in the reference image. However, a severe semantic and topical dissonance is introduced by the search result's title. The title claims the content is related to "Wwe 2K25" and the wrestler "Roman Reigns," a subject from the sports entertainment and wrestling simulation genre. This is in complete contradiction with the visual evidence, which portrays a character from a high-fantasy setting. This stark mismatch between the visual content and its textual description renders the entire result misleading. Therefore, despite the perfect visual correspondence between the images, the search result as a whole is highly irrelevant and deceptive, as the title fundamentally misrepresents the subject matter of the visuals, likely functioning as clickbait.

## Result 5

**web_info:** The provided webpage content does not contain information about the publisher of the game "Clair Obscur: Expedition 33." While the page is for a YouTube video related to this game and mentions "Sandfall Interactive" in the context of a "Behind the Voices" video, it does not name a publisher. Therefore, it is not possible to identify other games released by the publisher in 2024 or any details about their white-label publishing services based on the given text.

**related_info:** The analysis reveals a very high degree of relevance between the reference content and the search result. Visually, the character depicted in the reference image and the thumbnail are identical, sharing the same dark hairstyle, beard, facial structure, and dark, multi-layered attire. The somber, low-light aesthetic and cinematic quality are also consistent across both images, suggesting they originate from the same source. Semantically, the title "Clair Obscur: Expedition 33" identifies the source as a specific video game, which perfectly aligns with the high-fidelity, computer-generated appearance of the character and environment. The topic overlap is therefore complete, as both the reference and the search result pertain to the same character from the same video game. Given these strong visual and contextual connections, the search result is exceptionally relevant, as it appears to be a video clip featuring the very same character highlighted in the reference image.

## Result 6

**web_info:** Based on the provided web page, the publisher of the game "Clair Obscur: Expedition 33" is Kepler Interactive. However, the content does not contain any information about other games released by Kepler Interactive in 2024, nor does it mention the company's white-label publishing service.

**related_info:** The reference image and the search result demonstrate an extremely high degree of relevance. The visual features show a perfect match, as the male character in the reference image is clearly the same individual featured on the left side of the search thumbnail, identifiable by his hairstyle, facial structure, and distinctive dark outfit with shoulder details. This strong visual consistency immediately establishes a direct link. Semantically, the search result's title, "Clair Obscur: Expedition 33," and the text overlay on the thumbnail explicitly name the video game from which the character originates. The topics are not just overlapping but are identical, with the reference being a character from a specific game and the result being a gameplay video of that same game. Therefore, considering the identical core character and the explicit textual confirmation of the game's title, the search result is unequivocally and highly relevant to the reference image, representing a direct and specific piece of content about the source material.

## Result 7
**web_info:** The provided web page content is a "404 Page not found" error from the Douyin website. It does not contain any images or information related to a specific game, its publisher, or a list of games released in 2024. Therefore, it is not possible to answer the question based on the given text.
**related_info:** (Chinese results abbreviated)
## Result 8
**web_info:** Based on the provided web page content, it is not possible to answer your question. The text consists of a generic YouTube page layout and does not contain any specific information about the game, its publisher, or a list of games released in 2024.
**related_info:** The analysis reveals a very strong connection between the reference image and the search result. Visually, the core element, a male character with dark hair, a beard, and a distinctive layered dark outfit with gold accents, is perfectly matched between the reference and the thumbnail. The reference provides a closer view of the character, while the thumbnail shows the same individual in a wider scene, confirming a direct correspondence in character design, attire, and the overall dark, cinematic art style. Semantically, the search result's title, "WHEN ONE FALLS WE CONTINUE – Clair Obscur Expedition 33 Ep. 2," provides the specific context, identifying the source as the video game "Clair Obscur: Expedition 33." This textual information solidifies the connection, confirming that both images depict content from the same game. The topic overlap is therefore complete, as the search result is a gameplay video that explicitly features the character shown in the reference image. Consequently, the search result is exceptionally relevant to the reference content, representing a direct and specific match from the exact same source material.
## Result 9
**web_info:** Based on the provided web page, the publisher of the game "Expedition 33" is Hello Games. The content is a review that details this specific game's story, strategic combat system, and character customization. However, the text does not contain any information regarding other games released by Hello Games in 2024, nor does it mention a white-label publishing service. The page focuses exclusively on the features and description of "Expedition 33".
**related_info:** (Chinese results abbreviated)
## Result 10
**web_info:** Based on the provided web page, the publisher of the game *Clair Obscur: Expedition 33* is Kepler Interactive, and the developer is Sandfall Interactive. The game is planned for release on Xbox, PlayStation, and PC via Steam.

However, the webpage content, which is a YouTube Shorts video page for a review of this specific game, does not contain any information about other games released by Kepler Interactive in 2024. Additionally, there is no mention of the company's white-label publishing service or a list of titles released under it. Therefore, the provided text does not contain the information needed to answer your question.
**related_info:** The analysis indicates a very high degree of relevance between the reference image and the search result. Visually, the character in the reference image is a perfect match for the male character prominently displayed in the search result's thumbnail. The distinctive features, including the dark hair, beard, and specific style of the layered coat, are identical in both images. Furthermore, the dim, atmospheric lighting and the overall graphical art style are consistent, strongly suggesting that both images originate from the same source. The semantic context provided by the title, "Clair Obscur: Expedition 33," and the "#gaming" tag solidifies this connection by identifying the specific video game from which these visuals are taken. The reference image is essentially a cropped or focused shot of the same character within the same game world depicted in the thumbnail. Consequently, as the visual subject matter is identical and the textual information explicitly names the source content, the search result is directly and unequivocally relevant to the provided reference.
`</image_search_result>`
**assistant:**

**Tool Call:** `text_search`

`{`

```
  "queries": [
    "Kepler Interactive games released in 2024",
    "Kepler Interactive white-label publishing service",
    "Clair Obscur: Expedition 33 publisher"
  ]
}
```

**tool:**

`<text_search_result>`
## Result 11
**web_info:** Based on the provided web content, the Steam search page for the publisher Kepler Interactive shows "0 results match your search." Consequently, the page does not list any games from this publisher, and therefore contains no information regarding any titles released in 2024.
## Result 12
**web_info:** Based on the provided web content, Kepler Interactive released the game **Ultros** on February 13, 2024. It is an indie adventure game available on PlayStation 5, PlayStation 4, and PC (via Steam and Epic). In *Ultros*, players awaken on a giant, cosmic, space-drifting entity called The Sarcophagus and are trapped in a time loop. The game is a psychedelic sci-fi journey that explores themes of life, death, and karmic cycles. Its gameplay contrasts brutal, close-quarters combat with contemplative gardening, and a unique loop mechanic allows players to restart the story to unlock new areas and abilities.
The webpage also highlights **Pacific Drive** as being "out now," indicating a recent release. It is a survival game where the player must navigate the anomaly-filled Olympic Exclusion Zone with a car as their only lifeline.
No other games listed on the page are explicitly mentioned as having a 2024 release date in the provided text.
## Result 13
**web_info:** Based on the webpage content, Kepler Interactive released the following games in 2024:
* **Pacific Drive**: Released on February 22, 2024, for PC and PlayStation 5. * **Flintlock: The Siege of Dawn**: Released on July 18, 2024, for PlayStation 5, Xbox Series X—S, and PC. * **Cat Quest III**: Released on August 8, 2024, for platforms including iPhone, Macintosh, PC, Xbox Series X—S, Xbox One, and PlayStation 5.
## Result 14
**web_info:** Based on the provided content, Kepler Interactive released several games in 2024 under its main publishing label and its white-label service, Kepler Ghost.
**Games Published by Kepler Interactive in 2024:** * _Ultros_: Developed by Hadoque for macOS, Windows, Nintendo Switch, PlayStation 4/5, and Xbox Series X/S. * _Pacific Drive_: Developed by Ironwood Studios for Windows and PlayStation 5. * _Cat Quest III_: Developed by The Gentlebros for Windows, Nintendo Switch, PlayStation 4/5, and Xbox One/Series X/S. * _Flintlock: The Siege of Dawn_: Developed by A44 for Windows, PlayStation 4/5, and Xbox One/Series X/S.
**Games Published via Kepler Ghost in 2024:** * _Windblown_: Developed by Motion Twin. * _Unrailed 2: Back on Track_: Developed by Indoor Astronaut for Windows, macOS, and Linux. * _Dark Hours_: Developed by Piece of Cake Studios for Windows.
## Result 15
**web_info:** The provided web content does not contain information about Kepler Interactive games released in 2024. The content is a security notification from Cloudflare for the website gg.deals. It indicates that access to the target page was blocked with a "403 Forbidden" error, requiring a security check to verify the user is human before proceeding.
## Result 16
**web_info:** According to the provided text, Kepler Interactive launched a white-label publishing service in 2023 called Kepler Ghost. This service serves as an imprint for the company. The launch of Kepler Ghost is part of the company's strategy to scale its overall output, in addition to bringing new studios into its core co-ownership group. In 2024, Kepler released

five titles through either publishing agreements or its Kepler Ghost imprint: *Pacific Drive*, *Ultros*, *Dark Hours*, *Unrailed 2: Back on Track*, and the early access release of *Windblown*.
## Result 17
**web_info:** In 2023, Kepler Interactive launched Kepler Ghost, a white-label publishing service designed to provide marketing and publishing support for game developers. The first video game project to utilize this program was _Eternights_ by Studio Sai. Other titles published under the Kepler Ghost label include _En Garde!_, _Apocalypse Party_, _Battle Shapers_, _Windblown_, and _Unrailed 2: Back on Track_.
## Result 18
**web_info:** Kepler Interactive offers a publishing service named Kepler Ghost, designed for developers with fully funded projects seeking expert marketing and publishing support. This service provides true AA & AAA publishing expertise, allowing studios to build their own brand and retain control of their marketing. It is presented as offering better terms than a standard publishing partner, with a base revenue share of 15%.
Kepler Ghost's support covers seven key areas: * **Brand:** Creating a strong, recognizable identity for the game. * **Digital:** Maximizing potential through targeted online marketing, distribution, and sales. * **Communications:** Managing public relations and social media to build a loyal community. * **Relationship:** Leveraging Kepler's industry network to forge valuable partnerships. * **Data:** Using analytics to provide actionable insights for game optimization. * **Business Development:** Identifying and executing growth strategies. * **Platform:** Managing the technical requirements to launch and maintain the game across various platforms.
Developers interested in this service can get in touch via an inbound form on the Kepler Interactive website.
## Result 19
**web_info:** Kepler Interactive has launched a new white-label publishing service for a curated set of games and studios. This service provides developers access to Kepler's in-house marketing and publishing talent for an industry-leading 15% base revenue share.
The primary goal of this service is to empower developers to remain self-published, allowing them to build their own brand and community with professional support.
Projects already utilizing this white-label service include *Eternights* from Studio Sai and the unannounced debut game from Hadoque, a Swedish developer led by El Huervo, an artist known for his work on the *Hotline Miami* series.
## Result 20
**web_info:** Based on the provided web page, there is no mention of a "white-label publishing service."
Instead, the content describes Kepler Interactive as a global publisher and game developer built on a unique co-ownership model. This model provides studios with a stake in the business and a leadership role in strategic decision-making, while ensuring they maintain their creative independence. The goal is to enable them to create fun and commercially successful games.
Additionally, Kepler Interactive runs Kowloon, a gaming fund that supports developers across the world. The company also has a submission process for developers who wish to pitch their games.
## Result 21
**web_info:** According to the provided web page, the publisher for *Clair Obscur: Expedition 33* is Kepler Interactive.
The developer, Sandfall Interactive, secured a partnership and funding from the publisher Kepler Interactive in early 2023. This deal was instrumental in expanding the development team and securing professional voice actors.
For the physical version of the game, distribution was handled by Maximum Entertainment in North America and Bandai Namco Entertainment in Europe.
## Result 22
**web_info:** According to the copyright information on the webpage, *Clair Obscur: Expedition 33* is published by Kepler Interactive Limited. The game was developed by Sandfall Interactive SAS.
## Result 23

**web_info:** Based on the provided webpage, the publisher of *Clair Obscur: Expedition 33* is Kepler Interactive. The game is the debut title from the developer Sandfall Interactive.
*Clair Obscur: Expedition 33* is a turn-based RPG with real-time combat mechanics, built on Unreal Engine 5. Set in a fantasy world inspired by Belle Époque France, the story follows Gustave, Maelle, and their fellow Expeditioners on a final mission. They have one year to destroy the "Paintress," a mysterious entity who, once a year, paints a number on her monolith, causing everyone of that age to vanish. The game features an active combat system allowing players to dodge, parry, and chain combos in real-time. It will be released on Xbox Series X/S, PlayStation 5, and PC (Steam and Epic Games Store).
## Result 24
**web_info:** According to the webpage, *Clair Obscur: Expedition 33* is published by Kepler Interactive. The game was developed by Sandfall Interactive.
## Result 25
**web_info:** Clair Obscur: Expedition 33 is published by Kepler Interactive, a global publisher and game developer with a unique co-ownership model. The game is the first title from the developer Sandfall Interactive.
It is a turn-based RPG developed on Unreal Engine 5 for new-gen consoles and PC. The game is described as a single-player, narrative-driven adventure set in an original world that blends fantasy with European "Belle Époque" art influences.
`</text_search_result>`
**assistant:**

`<answer>`The game shown in the image is *Clair Obscur: Expedition 33* [21, 22, 23, 24, 25]. The publisher of this game is Kepler Interactive [21, 22, 23, 24, 25].
Kepler Interactive has a white-label publishing service called Kepler Ghost [14, 16, 17, 18, 19]. Excluding the games released through this service, the games published by Kepler Interactive in 2024 are: * *Ultros*: An indie adventure game released on February 13, 2024 [12, 14]. * *Pacific Drive*: A survival game released on February 22, 2024 [12, 13, 14]. * *Flintlock: The Siege of Dawn*: An action game released on July 18, 2024 [13, 14]. * *Cat Quest III*: An action RPG released on August 8, 2024 [13, 14].`</answer>`;

**Task:**
Which city is this?

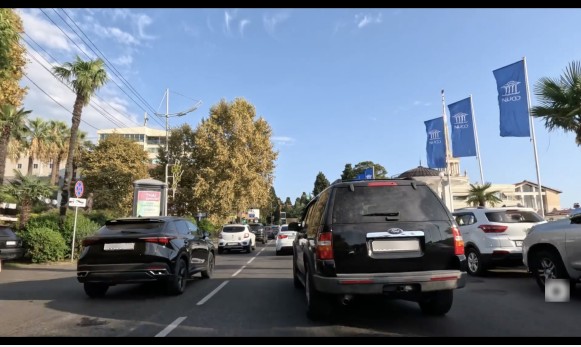

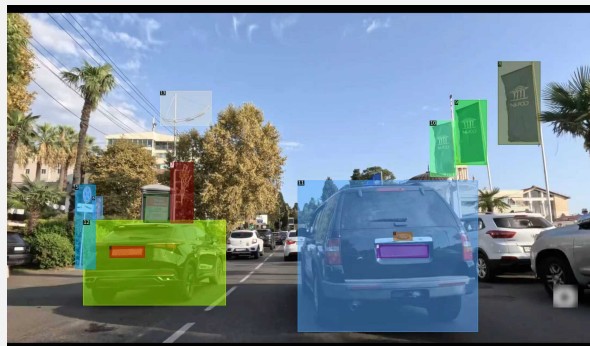

**assistant:**

**Tool Call:** `image_zoom_in_tool`

```
{
  "bbox_index": 8,
  "image_name": "he5murslfz.png",
  "label": "flag"
}
```

**tool:**

The cropped sub-image is provided in the next user round.

**user:**

`<zoom_in_image_name>`he5murslfz_round_1_crop_1.png`</zoom_in_image_name>`;
`<image_zoom_in_result>`

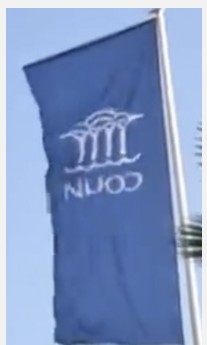

`</image_zoom_in_result>`
**assistant:**

**Tool Call:** `image_search`

```
{
  "image_name": "he5murslfz_round_1_crop_1.png"
}
```

**tool:**

`<image_search_result>`
## Result 1
web_info: According to the provided webpage, the headquarters of the Institute for Information Industry is located in Taipei City, Taiwan. Specifically, its main office is in the Technology Building in the Daan District of Taipei. The organization also has several other offices and departments located in different buildings throughout Taipei City, as well as locations in Nantou City and Kaohsiung City. Additionally, it maintains overseas offices in Tokyo, Japan, and Chennai, India.

## Result 2
web_info: The provided web page content does not contain any information about a city. The text is a security notification from Cloudflare for the website 'www.cake.me'. It indicates that the system was verifying the user's connection to review its security before proceeding. The content also includes a warning about a "403 Forbidden" error from the target URL, suggesting a potential access issue with the page.
## Result 3
web_info: The provided web page, which describes a stock video of the World Intellectual Property Organization (WIPO) logo, does not explicitly state a specific city as the location. However, "Ankara" is listed as one of the image keywords associated with the video.
## Result 4
web_info: The main article discusses a drug approval that occurred in Japan. However, the website's footer provides information indicating it is operated by a company based in Shanghai.
## Result 5
web_info: According to the copyright information on the webpage, the company that owns the XIBERIA brand is Dongguan Desheng Industrial Co., Ltd., which is located in Dongguan City.
## Result 6
web_info: The provided webpage content does not mention a specific city. The page is a product listing for a "Union of South American Nations Vertical Garden Flag" from the seller Pixelforma.
The flag is described as a durable, decorative item for outdoor spaces, made of weather-resistant polyester with double-sided printing for visibility from all angles. Its dimensions are 32 x 47.5 cm. The product includes only the flag; the holder is sold separately.
The regular price is listed as $21.00, but the item is currently unavailable and on backorder. The page also provides information on processing times (3 to 5 business days) and notes that delivery times are estimated based on the customer's location.
## Result 7
web_info: Based on the webpage's footer, the company operating this website, MedSci, is located in Shanghai, China. This is indicated by several pieces of information: (Chinese results abbreviated).
## Result 8
web_info: Based on the provided web content, the Birch Wathen Lenox School (BWL) is located in New York City (NYC). This is indicated in an Instagram post introducing the school's new clarinet teacher, Jeff Nichols. The caption for his introduction states, "Since moving to NYC post-college, he's dedicated his career to both teaching and performing."
## Result 9
web_info: The provided web page does not focus on a specific city. It is a page on the Japanese website APPLION that recommends various "footprint tracking" apps for the iPhone. The developers of the listed applications are based in several different cities, including Foshan, Shanghai, Xi'an, Chengdu, Wuhan, and Guangzhou.
## Result 10
web_info: Based on the provided web page content, there is no information available to identify a specific city. The page is a standard Facebook interface showing a "You're Temporarily Blocked" message, indicating that a feature was used too quickly. The content also includes a login form, options to create a new account or recover a forgotten one, a list of languages, and numerous links to various Meta products (like Messenger, Instagram, Meta Quest), services, and policies (such as Privacy Policy, Terms, and Help Center).
`</image_search_result>`
**assistant:**

**Tool Call:** `text_search` *(Translated from Russian)*

```
{
  "queries": [
    "NIKOO logo blue flag",
    "NIKOO company",
```

```
    "NIKOO city"
  ]
}
```

**tool:**

```
<text_search_result>
```
## Result 11
web_info:
Nikao Technology is a company providing innovative digital solutions and consulting services, specializing in wealth management and investment. It offers expert digital consulting and customized strategies aimed at driving business growth and success.
Key areas of expertise include: * **Product launch:** Development and launch of savings and pension products for banks and retail clients. * **Platforms:** Creation of omnichannel web and mobile platforms for investment and product management. * **Migrations:** Managing complex client and asset migrations between different investment platforms, including FNZ and GBST. * **Artificial intelligence:** Using AI for market analysis, document processing, and creating financial data stores.
The company collaborates with financial institutions and asset managers, focusing on innovation and long-term partnerships to provide cutting-edge solutions.
## Result 12
web_info: (Translated from Russian) Nikoyo (HK) Ltd., founded in Hong Kong in 1985, is part of the Nikoyo (Holdings) Ltd. The company manages commercial, financial, and logistics operations, signing contracts with global partners for product distribution and providing platforms for sales, software development, and technical support.
Nikoyo offers comprehensive IT solutions, including business process management (BPM), cloud services, cybersecurity, digital transformation, infrastructure, intelligent information management, IoT, and AI, as well as robotic process automation (RPA). The company also provides professional IT services: software development, managed services, consulting, and training.
Key development milestones include obtaining exclusive agency rights for Fujitsu products in 1986, launching NetApp and EMC products on the mainland of China in the 1990s, and developing its own content management system "DocuMan uContent" in 2002. Over the years, Nikoyo has implemented numerous projects for clients from various sectors, including government agencies, global banks, and transportation companies. The company has ISO certifications and collaborates with leading technology companies such as Cisco, Dell, IBM, Microsoft, NetApp, and VMware.
## Result 13
web_info: (Translated from Russian) Nikko Co., Ltd. (Nikko Co., Ltd.) is a Japanese company founded on August 13, 1919. The main office is located in the city of Akaashi, the Hogo Prefecture, and there is also an office in Tokyo.
**Main business:** The company specializes in producing asphalt mixing and concrete mixing equipment, belt conveyor systems, as well as equipment for protecting the environment, including plastic recycling equipment, beverage containers, and soil reclamation. It also develops control panels and software.
**Key data:** * **Capital:** 9.197 billion yen. * **Net sales:** 49.1 billion yen (as of March 31, 2025). * **Employees:** 1,397 people. * **Certification:** ISO-9001 and ISO-14001.
**History:** Originally founded as Nihon Kogu Product Co., Ltd., the company started with producing shovels. In 1958, it produced its first asphalt mixing plant, and in 1968, it changed its name to Nikko Co., Ltd. In August 2019, the company celebrated its 100th anniversary. Nikko has international presence, including a subsidiary in Shanghai and sales offices worldwide.
## Result 14
web_info: (Translated from Russian) CNOOC Limited (CNOOC Limited) is the largest producer of offshore crude oil and natural gas in China. It is the main subsidiary of the state-owned China National Offshore Oil Corporation (CNOOC).

The company was registered in Hong Kong in August 1999 and listed on the Hong Kong Stock Exchange from February 2001. Previously, it was also listed on the New York Stock Exchange (2001-2021) and Toronto Stock Exchange (2013-2021).

The main areas of activity of CNOOC Limited in China are the Bohai Sea, the South China Sea, and the East China Sea. On the international level, the company has oil and gas assets in Asia, Africa, North America, and Oceania.

CNOOC Limited is part of the "Big Three" Chinese oil companies alongside CNPC (specializing in onshore production) and Sinopec (specializing in refining and marketing), although these differences have diminished, and companies compete in all sectors.

## Result 15

web_info: (Translated from Russian) The provided web page content does not contain information about the company "NIKOO".

The text is a system error message from Google/YouTube. It states that access to the target URL was blocked due to the detection of unusual traffic from the user's computer network, which may violate the Terms of Use. Possible causes of such traffic are malicious software, browser plugins, automated scripts, or the use of a common IP address with another device, violating the rules.

## Result 16

web_info: (Translated from Russian) Nikko (Japanese, meaning "sunlight") is a city in Japan, located in the Totigi Prefecture, 115 kilometers north of Tokyo. As of 2020, its population was 80,239 people. Nikko ranks third in Japan by municipal area (1,449.83 square kilometers), with 90% of its territory consisting of mountains.

The city was founded in 782. It became an important religious and postal center during the Edo period, especially after the construction of the Tosho-gu Shrine in 1617. In 1954, Nikko was granted city status. The economy includes food processing, aluminum rolling, and copper foil production.

Nikko is a popular resort and major tourist center. Its main attractions are the Tosho-gu Shrine and Futarasan Shrine, as well as the Rinno-ji Buddhist temple, which has been included in the UNESCO World Heritage List since 1999. Natural features such as Lake Tsudzendo, Kegon Waterfall, and Senjogahara Bog are part of the Nikko National Park. The city is named after the asteroid (1185) Nikko.

## Result 17

web_info: (Translated from Russian) Nikko is a region north of Tokyo, known for its beautiful mountainous landscapes and hot springs. The "Travel to Tokyo and Nikoo" tour includes a two-day visit to this area.

The program for Nikko includes: * **Natural attractions:** Visiting the Kegon Waterfall (97 meters high, one of the three most famous waterfalls in Japan), Lake Tsudzendo, and descending the famous zigzag road Iroha. * **Cultural and historical sites:** Visiting the Tosho-gu Shrine, the mausoleum of the great Tokugawa Ieyasu, a XVII-century complex that is a UNESCO World Heritage Site and known for its rich decoration. * **Entertainment:** Visiting the Edo-mura theme park, where the life of Japan during the XVII-XIX centuries is recreated with ninja and samurai shows. * **Accommodation:** Staying in a traditional Japanese hotel (ryokan) at Lake Tsudzendo with a Japanese-style dinner and visiting the natural hot springs.

After the tour of Nikko, it is planned to return to Tokyo on the 10th day.

## Result 18

web_info: (Translated from Russian) Ningguo is a county-level city in the Sunchang District of Anhui Province, China. Its area is 2,466.9 square kilometers, and its population as of 2020 was 384,644 people.

The history of Ningguo began in 208 when it was founded as a city. Over the centuries, its administrative status changed multiple times, it was a county, a prefecture, and a region. In 1997, the county of Ningguo was transformed into a county-level city, and in 2000, it became part of the newly established Sunchang District.

Ningguo is administratively divided into 6 street committees, 8 towns, 4 townships, and 1 national township.

## Result 19

web_info: (Translated from Russian) The Hotel Nikoo is located in the city of Meshhed, Iran. It is in the center of the city, 1 kilometer from the center, at Imam Reza Street, Baite Makdis Square.

This is a one-star budget hotel offering affordable accommodation. Its key advantages are its convenient location near main attractions, including the Imam Reza Shrine, and its view of it. The hotel is suitable for both tourists and business travelers.

The hotel offers comfortable rooms and quality amenities such as 24-hour reception, free breakfast, free Wi-Fi, and private bathrooms. The hotel is known for its high level of service and hospitality of the staff.

The check-in time is from 14:00, and the check-out time is before 12:00. It should be noted that the hotel does not accept international credit/debit cards.

## Result 20

web_info: (Translated from Russian) The provided Facebook web page does not contain information related to the request "NIKOO city".

The content of the page is a notification of a temporary block of the user. The reason for the block is the misuse of the function due to too fast actions. The page also contains standard elements for logging in, creating a new account, recovering access, and selecting the interface language.

```
</text_search_result>
```
**assistant:**

**Tool Call:** `image_zoom_in_tool`

```
{
  "bbox_index": 4,
  "image_name": "he5murslfz.png",
  "label": "sign"
}
```

**Tool Call:** `image_zoom_in_tool`

```
{
  "bbox_index": 7,
  "image_name": "he5murslfz.png",
  "label": "sign"
}
```

**tool:**

The cropped sub-image is provided in the next user round.
**user:**

`<zoom_in_image_name>`**he5murslfz_round_4_crop_1.png**`</zoom_in_image_name>`;
`<image_zoom_in_result>`

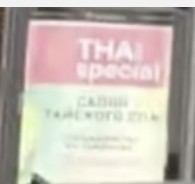

`</image_zoom_in_result>`
**tool:**

The cropped sub-image is provided in the next user round.
**user:**

`<zoom_in_image_name>`**he5murslfz_round_4_crop_2.png**`</zoom_in_image_name>`;
`<image_zoom_in_result>`

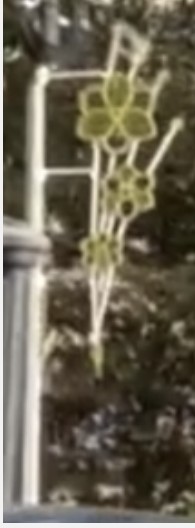

```
</image_zoom_in_result>
```
**assistant:**

**Tool Call:** `text_search`

```
{
  "queries": [
    "NIKOO Sochi",
    "NIKOO real estate Sochi",
    "builder NIKOO Sochi"
  ]
}
```

**tool:**

```
<text_search_result>
```
## Result 21
web_info: (Russian results abbreviated)

...
## Result 34
web_info: (Russian results abbreviated)
```
</text_search_result>
```
**assistant:**

**Thinking Process**
1. **Initial Analysis of the User's Question:** The user wants to identify the city shown in the image.
2. **Visual Clues from the Image:** * The environment features palm trees and deciduous trees, suggesting a temperate or subtropical climate. * The cars and road infrastructure look modern. * There are several flags and signs with text. The most prominent are the blue flags (e.g., bbox 8, 9, 10). The text on them appears to be in the Cyrillic alphabet.
3. **Strategy Formulation:** * **Step 1:** Identify the text on the most prominent objects. The blue flags (bbox 8) are the clearest starting point. I will zoom in on one to read the text accurately. * **Step 2:** Once the text is identified, I will use a text search to find out what organization it represents and where it is located. The Cyrillic script suggests searching in Russian will be most effective. * **Step 3:** To confirm the location, I will look for other textual clues in the image. The signs at bbox 4 and 7 could provide additional information. I will zoom in on them as well. * **Step 4:** Based on the information gathered, I will refine my search queries, likely by combining the identified name with potential city names that

fit the visual context (a Russian-speaking city with palm trees). * **Step 5:** Synthesize all the evidence to make a final conclusion about the city.

4. **Execution of the Plan:** * **Round 1 (Zoom):** I started by zooming in on the blue flag (bbox 8). The resulting image clearly shows the text and a logo. * **Round 2 (Search):** I performed an image search on the cropped flag, but the results were generic and unhelpful. I then conducted a text search in Russian. The results pointed to various companies named "Nikko," "Nikoyo," or "Nikoo" in Japan, Hong Kong, and Iran, none of which fit the visual context of a Russian-speaking city. This indicated that `russian-1` might be a local company, and I needed more context. * **Round 3 (Zoom):** To gather more information, I zoomed in on the signs at bbox 4 and 7. The sign at bbox 4 was a billboard. The zoomed-in image revealed the text ("Thai Spa Salon") in Russian. This confirmed the language and that the location is in a Russian-speaking country. The palm trees strongly suggested a southern city, like Sochi. * **Round 4 (Refined Search):** Armed with the name `russian-1` and a strong hypothesis that the city is Sochi, I performed a targeted search using queries like "russian-2" and "russian-3" (NIKOO real estate Sochi). * **Analysis of Search Results:** The search results returned information about several companies named `russian-4` (a very close match to `russian-1`) in Sochi. Specifically, search results [25, 30, 31, 33] mention that `russian-4` was a company involved in real estate and construction in Sochi. Although the records indicate these specific entities were liquidated, their existence strongly links the name on the flags to this city.

5. **Conclusion:** The combination of visual evidence (palm trees, Russian-language signs) and the results of the targeted text search, which linked the company name `russian-4` or `russian-1` to real estate and construction businesses in Sochi, confirms that the city in the image is Sochi, Russia.

**¡answer¿** Based on the visual evidence and search results, the city in the image is **Sochi, Russia**.

Here is a step-by-step summary of how this conclusion was reached: 1. Initially, I identified prominent blue flags with the text `russian-1` written in the Cyrillic alphabet. 2. I then zoomed in on another sign in the image (a billboard), which had the Russian text ("Thai Spa Salon"). This, combined with the presence of palm trees, strongly suggested the location is a southern Russian city. 3. A targeted search for the company `russian-1` in Sochi revealed information about a company named `russian-1`, which was involved in real estate and construction in the city [25, 30, 33]. While some records show the company was liquidated, the presence of these flags confirms a connection between the brand and the city of Sochi. **¡/answer¿**

## L  PROMPTS

**System Prompt**

You are a question–answering expert. You are skilled in flexibly using search tools and image processing tools to answer users' question. The search tools include image search and text search. The image processing tools include object grounding. In the first round of user–submitted content, there is an image containing object bounding boxes and their corresponding index numbers, which is denoted as the annotation image. In each round, users will send you text or images, and you need to:
1. Break down the question based on its complexity, analyze what information is needed to answer it, and then use multiple rounds of tool calls to respond to the user's question.
2. For complex questions, pay attention to combining the initial analysis strategy and planning during the reasoning process, gradually obtaining relevant information.
3. For text search, you can generate 3 to 5 queries at a time and send them to the search engine to improve retrieval efficiency; if the returned results are not as expected, you can try modifying the search queries to obtain more useful information.
4. Note that when you are uncertain about the image or certain concepts, try to use tools to get the answer unless you are very sure that the relevant content is correct.

5. Determine the retrieval strategy needed based on the question and the information directly obtainable from the image. If the target in the image is too small or unclear, you can crop the sub−image to obtain a clearer view. Specifically, you can **obtain the sub−image by requesting the index number of a specific subregion provided in the annotation image of the first round**. When the image search results are not ideal, make full use of the concepts and text information in the image to find the answer and relevant information to the question.

6. After each round of tool calls, you will receive the return results of the tool calls. You need to decide whether to continue calling tools and which tools to call.

7. Note that after each round of tool calls, fully analyze the current state based on the completed execution results, appropriately review the user's original question and reflect, determine whether the overall plan needs to be adjusted, and whether to continue calling tools.

8. When information is to be returned, historical information can already answer the user's question, first **summarize the overall thought process and the process of obtaining the answer to ensure accuracy, and check whether your answer indeed answers the user's question, avoiding irrelevant responses**, then output the answer, with the answer fragment written between <answer> and </answer>.

9. When outputting the answer, remember to include citations, using the '[quote_id]' format, where 'quote_id' corresponds to the numerical ID of the webpage.

10. For text search, each query must be **a keyword, phrase, or sentence**.

