# OpenReview forum: "MMSearch-Plus: Benchmarking Provenance-Aware Search for Multimodal Browsing Agents"
_ICLR.cc/2026/Conference — ICLR 2026 Poster_

### Official Review · Reviewer_sEzM · 2025-10-26

**Soundness:** 3
**Presentation:** 3
**Contribution:** 2
**Rating:** 6
**Confidence:** 4

**Summary:**

This paper introduces MMSearch-Plus, a multimodal browsing benchmark that requires models to perform iterative multimodal retrieval and cross-validation under noise. The benchmark is constructed using a spatial-temporal extrapolation curation pipeline that constructs 311 tasks requiring models to extract localized visual cues and extrapolate to out-of-image facts. The authors also proposed an agent framework with web search tools and a set-of-mark module for the proposed task. The paper further evaluates different closed-source and open-source models with the proposed framework on the proposed benchmark and provides error analysis.

**Strengths:**

1. The paper clearly identifies a real weakness in existing multimodal search benchmarks—tasks solvable with text-only heuristics—and systematically designs MMSearch-Plus to require true multimodal reasoning.

2. The paper proposed a suitable agent framework for the proposed benchmark and conducted comprehensive evaluations for different models with multiple search modes, providing a solid empirical foundation for further research.

3. The authors provide detailed error categorization, offering actionable insights into model weaknesses in long-horizon multimodal search.

**Weaknesses:**

1. The dataset size is relatively small, and all samples are generated through the same spatial-temporal extrapolation pipeline. This design choice may introduce human bias and constrain the diversity of reasoning patterns, leading to potentially predictable task structures. As a result, the benchmark might be vulnerable to overfitting or data-specific “hacks,” where a model trained on a small amount of similar data could achieve disproportionately high scores.

2. To better support interpretability and diagnostic analysis, it would be highly beneficial to annotate fine-grained evidence—for example, marking which visual regions are essential or sufficient for solving each task. Such annotations would allow for a deeper understanding of model failures and reasoning behaviors.

3. While the paper introduces zoom-in and cropping tools within the evaluation framework, it lacks experimental evidence demonstrating whether these tools are genuinely necessary or beneficial for specific task types, or whether current models can effectively leverage them.

4. Some tasks can be solved using the model’s internal knowledge rather than external retrieval, which weakens the benchmark’s diagnostic focus on search-based reasoning. Future iterations might consider constructing questions from recent or dynamically updated sources and designing a reusable pipeline to continuously refresh the benchmark content, ensuring long-term relevance.

5. The low human performance reported on MMSearch-Plus raises concerns about possible ambiguity or excessive difficulty in certain questions. This suggests that some samples may not have a clearly defined or uniquely inferable answer, limiting their reliability for model evaluation.

**Questions:**

Since the benchmark emphasizes iterative multimodal search and validation, the “easy” split includes samples that certain models can already answer correctly without any search steps. Why do the authors still keep these samples in the benchmark? Would it be more reasonable to have human annotators label the necessary reasoning or tool-use steps and then categorize difficulty based on that process?

---

> ### Author Response · Authors · 2025-11-20
>
> We thank the reviewer for their thoughtful comments and for recognizing our benchmark’s focus on genuine multimodal reasoning. We address the concerns below.
>
> **W1. Dataset size and diversity**
>
> Although STE is used only in the final difficulty-raising stage, the overall pipeline is more diverse than it may appear. We manually curate visually rich videos, author intuitive questions, and test whether state-of-the-art MLLMs can answer them without retrieval. Only when models succeed do we apply STE to introduce controlled spatial–temporal transformations. This yields nontrivial variants while preserving broad reasoning diversity.
>
> **W2. Fine-grained evidence annotations**
>
> Because our pipeline stores cached image-search results for all subregions produced during SoM-based segmentation, we can determine sufficiency by checking whether the search results for a specific subregion contain webpages that fully answer the question. However, identifying essential regions is nontrivial. Many tasks (e.g., geo-localization) rely on extremely subtle cues. Furthermore, multiple distinct reasoning paths may lead to the correct answer, making essentiality non-unique. We are labeling the key regions and plan to release all annotations within the next month.
>
> **W3. Necessity and usage of zoom/crop tools**
>
> Zooming improves performance by 1–3%. Models often use zoomed views to refine internal visual understanding rather than directly consuming cropped-search results—consistent with “think-with-image” behavior. Human annotators routinely relied on image search of cropped regions, confirming the tools’ real utility even if current models underutilize them.
>
> **W4. Internal knowledge vs. retrieval; benchmark refresh**
>
> We share the concern that some tasks may become solvable using internal model knowledge. This issue becomes more severe as time passes, since closed-source models continue to be trained with increasingly recent knowledge cutoffs. The only reliable way to prevent this is to update the benchmark with knowledge that is newer than what existing models have been trained on.
>
> Beyond manual curation, prior work such as WebSailor [1] has proposed methods for automatically generating QA pairs. For VQAs, BrowseComp-VL [2] similarly introduced a pipeline that includes selecting suitable images. However, to make BrowseComp-like questions genuinely multimodal, we must choose images that are sufficiently complex to require multi-round visual reasoning. Simply replacing a text entity with an image of that entity is insufficient—current MLLMs can easily translate the image back into text, reducing the task to a purely text-based browsing problem.
>
> To keep MMSearch-Plus relevant, we plan to pursue semi-automated multimodal data construction, add new tasks through author and community contributions, and periodically update the dataset to mitigate internal-knowledge shortcuts.
>
> **W5. Human performance and ambiguity**
>
> Our outsourced annotators were not domain experts, explaining cases where models outperform them even without retrieval. To reduce ambiguity, we include auxiliary images or hints when needed, with reference to the "checklist" metadata in MM-BrowseComp [3]. Newly added human trajectories and evidence regions (W2) provide additional validation; these will be released within a month.
>
> **Q1(a). Why keep VQAs solvable without search?**
>
> During construction, we removed questions answerable by GPT-4o/o3 without retrieval. Later closed-source model updates made some previously difficult items solvable, so we selected a subset of 239 tasks as MMSearch-Plus-lite, containing only questions unsolvable by all models without search (Appendix H). We will continue refreshing the dataset to suppress internal-knowledge shortcuts.
>
> **Q1(b). Difficulty defined by human trajectory length**
>
> We appreciate the suggestion. Once all ground-truth trajectories are collected, we will revise difficulty annotations based on the number and complexity of reasoning and tool-use steps required by human annotators.
>
> [1] WebSailor: Navigating Super-human Reasoning for Web Agent
>
> [2] WebWatcher: Breaking New Frontier of Vision-Language Deep Research Agent
>
> [3] MM-BrowseComp: A Comprehensive Benchmark for Multimodal Browsing Agents

---

> > ### Comment · Reviewer_sEzM · 2025-11-27
> > **Reply to authors's rebuttal**
> >
> > I thank the authors for their detailed response. I appreciate the clarifications on tool usage and the commitment to releasing key region annotations. While the benchmark's limited size still poses some risk regarding coverage and overfitting, I believe the rigorous curation pipeline and planned future updates outweigh this limitation. Consequently, I will maintain my positive rating.

---

### Official Review · Reviewer_SjaW · 2025-10-30

**Soundness:** 3
**Presentation:** 3
**Contribution:** 3
**Rating:** 6
**Confidence:** 3

**Summary:**

MMSearch-Plus is a multimodal searching benchmark with 311 tasks that forces multimodal reasoning by requiring agents to extract fine-grained, localized visual cues and propagate them through iterative image-text retrieval. The tasks also require provenance checks under retrieval noise to reach those "out-of-image" facts like events, dates, venues. The authors also introduce Spatial–Temporal Extrapolation to curate questions and provide an agent framework with a Set-of-Mark (SoM) zoom-and-retrieve module for searching. Results show that: even the best closed-source system reaches only 36.0% with full rollout, and SoM yields consistent gains up to +3.9 points; dominant failures involve missing relevant webpages and confusing visually similar events. The benchmark thus serves as a rigorous stress test and common yardstick for agentic MLLMs.

**Strengths:**

(1) The paper clearly identifies limitations in MMSearch and proposes a carefully designed dataset-curation process to construct a more challenging benchmark where information is intentionally hidden, thereby requiring genuine visual reasoning rather than shortcut cues.

(2) The experiments and evaluation are comprehensive and detailed, comparing cutting-edge open-source and proprietary models across four search modes and multiple task subsets.

(3) The analysis is thorough, offering interesting observations and insights alongside a well-reasoned error analysis.

**Weaknesses:**

(1) The average answer length is relatively short, suggesting that many items may be closer to MCQ-style or “single-point” questions; the benchmark may under-represent open-ended QA.

(2) While the benchmark is valuable for provenance-aware retrieval, it is less comprehensive for many real-world agent tasks like cross-site form/API interactions. This may bias the evaluation toward retrieval-and-verification strength while being less sensitive to interactive capabilities.

**Questions:**

(1) You note that models sometimes “zoom in without subsequently performing region-based retrieval.” Could you report the proportion of zoom actions that are followed by a subimage search, and quantify the marginal contribution of that step to final accuracy?

(2) Why does performance on the Easy subset decrease when moving from "image-only" search to the "full-rollout" setting? Is this due to distractor exposure or over-retrieval, and can you provide supporting diagnostics?

---

> ### Author Response · Authors · 2025-11-20
>
> We thank the reviewer for the thoughtful and constructive feedback, and we appreciate the positive assessment of our benchmark’s clarity, experimental depth, and analysis.  To address the concerns, we conducted additional analyses, and we have added both empirical results and clarifications in the revised paper.
>
> **W1. Short answer length and MCQ-style questions**
>
> The relatively short answer length is largely by design. Our goal is to evaluate research and reasoning quality in a consistent and automatable way, which is much easier when the final answers are “single-point” (often MCQ-style or short-form).
> Crucially, answer length is not a good proxy for task difficulty in our setting. The difficulty of a task in MMSearch-Plus is much better reflected by the complexity of the search + reasoning trajectory (e.g., the number and depth of image zooms, retrieval hops, and cross-page evidence aggregation) than by the surface length of the final answer string.
> We view “deep research” as naturally decomposable into two stages:
>
> - Research / information seeking: iteratively exploring the environment (search engine, webpages, images), conducting trial-and-error queries, verifying provenance, and resolving conflicts between sources.
> - Report writing / synthesis: organizing the gathered evidence into coherent, long-form text.
> MMSearch-Plus is explicitly designed to isolate and rigorously stress-test the first stage—the research and evidence-gathering capabilities of agentic MLLMs—without conflating this with their long-form generation or stylistic abilities. Our benchmark is therefore complementary to existing benchmarks that focus more directly on report-writing and long-form synthesis.
>
> **W2. Limited coverage of cross-site form/API interactions**
>
> We agree that many real-world agent tasks involve interactive capabilities such as cross-site form filling or calling external APIs. Our benchmark intentionally focuses on a narrower but fundamental capability: browsing-style, provenance-aware information seeking in a noisy open-web environment.
>
> The information-seeking behavior is a core component of many browsing agents, regardless of whether they eventually go on to fill forms, execute code, or call APIs. In other words, our benchmark is largely orthogonal to low-level “operation” or API-calling skills. We see extending MMSearch-Plus to richer action spaces—such as cross-site form/API interactions and code-based tool use—as a promising future direction, but we deliberately decouple this from our current goal.
>
> **Q1. The effect of the cropped image search**
>
> We analyzed the contribution of the “zoom-in-then-image-search” action sequence to final accuracy.
> - For Gemini-2.5-Pro, among the 39 questions that become newly correct when moving from full rollout to full rollout + SoM, 5 involve a trajectory where the model first zooms into a region and then performs an image search. This corresponds to 12.8% of the newly solved questions.
> - For the best-performing o3 model, among the 42 questions, only 1 case follows this “zoom then regional search” pattern, corresponding to 2.4%.
>
> This suggests that Gemini-2.5-Pro leverages fine-grained, region-based image search more frequently, whereas o3 often uses zoom primarily as a way to view the region more clearly, without subsequently invoking region-specific retrieval. We will add these statistics and clarify this behavioral difference in the camera-ready version.
>
> **Q2. The reasons for the performance drop on easy split**
>
> For o3, whose accuracy on the Easy split drops when moving from image-only search to the full-rollout setting, we conducted a targeted analysis. There are 23 questions for which the image-only setting is correct but the full-rollout setting is not, corresponding to 7.4% absolute accuracy on our 311-task benchmark.
>
> We randomly sampled 10 out of these 23 cases and manually inspected the trajectories:
> - In 9 out of 10 cases, the full-rollout trajectory never invoked image search at all. The model apparently believed it had fully understood the image and opted to rely solely on text search or prior knowledge, but this was suboptimal—the missing image search meant that critical, fine-grained visual cues were never retrieved.
> - In the remaining 1 case, the model did perform image search and retrieved relevant sources, but made a reasoning error in the final step, leading to an incorrect answer.
>
> These observations indicate that the drop in Easy-split accuracy is not primarily driven by over-retrieval or distractor exposure, but rather by underuse of image search in the more complex full-rollout setting—the model sometimes “skips” a necessary image-search step when it wrongly assumes that additional retrieval is unnecessary. We will include this diagnostic in the revision and clarify that the main failure mode is a decision-making issue (whether to use image search), rather than an inherent downside of the full-rollout setting itself.

---

### Official Review · Reviewer_WMkC · 2025-10-31

**Soundness:** 2
**Presentation:** 2
**Contribution:** 2
**Rating:** 4
**Confidence:** 4

**Summary:**

The authors propose a new VQA benchmark. To answer the questions, models should understand both spatial and temporal knowledge. The authors show that existing models perform poorly on the benchmark but achieve some improvement with set-of-mark prompting.

**Strengths:**

- The proposed benchmark, where questions require both visual and textual cues to be answered during web browsing, is interesting.
- The experiments are comprehensive, evaluating various models across different categories and difficulty levels.
- The proposed approach --- i.e., identifying subregions through set-of-marking and searching those regions on the website --- looks interesting and promising.

**Weaknesses:**

- Novelty of the benchmark: Evaluating a model’s spatial and external temporal knowledge has been widely studied in conventional VQA tasks (e.g., [1], [2]). I feel the authors simply extend these existing works to the web-browsing domain.

- The paper is somewhat difficult to follow, and its core contribution is not immediately clear at first glance.

- Since you call image/text search APIs for every SoM-defined subregion, the runtime could increase substantially, which may be too costly for real-world deployment. Do you have strategies to mitigate this overhead?

[1] Can Pre-trained Vision and Language Models Answer Visual Information-Seeking Questions?, EMNLP'23.

[2] Entity-Focused Dense Passage Retrieval for Outside-Knowledge Visual Question Answering, EMNLP'22.

[3] GQA: A New Dataset for Real-World Visual Reasoning and Compositional Question Answering, CVPR'19.

**Questions:**

- The text appearing within the image (i.e., scene text) could arguably be treated as both visual and textual information. Did you apply both image and text searches on this scene-text?

---

> ### Author Response · Authors · 2025-11-20
>
> We thank the reviewer for their thoughtful comments and constructive suggestions. We are encouraged by their recognition of the interesting nature of our benchmark, the breadth of our experiments, and the promise of searching visual subregions via SoM. To address the concerns, we conducted additional analyses, and we have added both empirical results and clarifications in the revised paper.
>
> **W1. Novelty of the benchmark and relation to prior work**
>
> We appreciate the pointer to prior VQA work on spatial and external knowledge. Our goal is not simply to “port” existing VQA benchmarks (e.g., InfoSeek, OK-VQA) to a web-browsing setting, but to study agentic multimodal search where the model must:
>
> 1. decide what to look at in an image (including subregions),
> 2. decide whether and how to call external tools (image vs text search),
> 3. solve tasks that are extremely unlikely to be answerable without online search.
>
> In addition, the training datasets for modern MLLMs often include “outside knowledge” that is later used as the retrieval corpus for benchmarks such as OK-VQA. As a result, retrieval in these benchmarks loses its intended meaning: models can rely on internally memorized information rather than retrieving it from an external source. We will clarify these distinctions in the revised version by explicitly situating our benchmark in relation to [1–3] in the related work section (Section 6, Appendix C and D).
>
> **W2. Clarity of the core contribution and presentation**
>
> We acknowledge that the core contribution may not have been immediately clear. In the revision,
> 1. We streamline the introduction to clearly list our contributions (benchmark design, SoM-based spatial grounding for search, and agent analysis)
> 2. A running example was added that follows a single task end-to-end.
> 3. Section 2 was tightened to separate benchmark construction from the agent framework more clearly.
> 4. We refine the writing of the entire paper.
>
> We believe these changes will make the paper easier to follow.
>
> **W3. Runtime and cost of SoM-based search**
> - Although SoM provides more tool-call possibilities, the agent has the freedom to choose whether to use it or not.
> - Tool calls are globally capped and typically small. Our agent pipeline limits the total number of tool calls (of any type) to a maximum of 20 per trajectory.
> - SoM is as a compromise for grounding, not an end-state design. We agree that in a fully general setting, an ideal agent would freely generate its own bounding boxes instead of relying on a fixed set from an external expert. Our SoM mechanism is a pragmatic compromise: current models often struggle to propose precise, semantically meaningful boxes on their own, so we provide a fixed set of candidate regions from an external “expert” detector. From this perspective, introducing SoM reduces the search space of possible regions compared to unconstrained box generation.
>
> **Q1. Image search vs. text search on scene text**
> The reviewer insightfully notes that scene text can be seen as both visual and textual information, and asks whether we apply both image and text searches. In our framework, the answer is: we do not hard-code a strategy; the agent chooses.
> Given an image that contains scene text, the model has (at least) two natural options:
> 1. Use its own OCR ability and call text search.
> - The model first “reads” the text from the image (using its built-in OCR capability) and then calls the text-search API with the extracted string as the query.
> 2. Use image search on the subregion containing text.
> - If the model is uncertain about the exact text or wants to exploit visual details (font, layout, background), it can instead call the image-search API on the SoM subregion that contains the text. A strong image-search engine may internally perform OCR and retrieve relevant pages, effectively mimicking the first strategy while also leveraging additional visual cues.
>
> To better understand how models behave in practice, we conducted an additional analysis focusing on tasks that contain scene text:
>
> - We first detected scene text and its bounding boxes with DeepSeek-OCR.
> - We then randomly sampled 10 tasks with detected scene text and manually inspected the full trajectories under the “full rollout + SoM search” setting.
> - For o3, in 8 of 10 trajectories the model used only text search, converting the scene text into textual queries. The remaining 2 trajectories contained no tool calls at all.
> - For Gemini-2.5-Pro, in 7 of 10 trajectories the model again used only text search. In 1 trajectory, the agent combined full-image search and text search, and in the remaining 2 trajectories it made no tool calls.
>
> These observations suggest that, at least on this sample, models overwhelmingly prefer to treat scene text as textual evidence and use text search, occasionally supplementing it with image search. We will include this analysis and a clearer explanation of the agent’s tool-choice mechanism in the revised version.

---

### Official Review · Reviewer_xCaP · 2025-11-01

**Soundness:** 3
**Presentation:** 4
**Contribution:** 3
**Rating:** 6
**Confidence:** 3

**Summary:**

This paper proposes a new benchmark called MMSearch-Plus (311 tasks) designed to force true multimodal reasoning by requiring Fine-grained visual cue extraction, search under retrieval noise, and multi-step visual–textual cross-validation. This addresses the drawbacks of existing multimodal browsing benchmarks, such as MMSearch, in which many tasks can be solved by text-only reasoning. It also contributes a
model-agnostic agent framework with standard browsing tools and a set of mark module, which lets the agent place marks, crop subregions, and launch targeted image/text searches. The authors show that even the strongest MLLMs do not perform well on this benchmark.

**Strengths:**

1. The core design decisions, such as requiring fine-grained, exhaustive visual reasoning for answering the question make sense.

**Weaknesses:**

1. The authors did not include statistics or any reference to how many questions in MMSearch are solvable by text-only browsing.

**Questions:**

How many questions in MMSearch are solvable by text-only browsing? This is an important motivation for the current benchmark.

---

> ### Author Response · Authors · 2025-11-20
>
> We appreciate the reviewer’s positive assessment of the core design decisions behind MMSearch-Plus, particularly the benchmark’s emphasis on fine-grained, exhaustive visual reasoning and its necessity for real multimodal “deep research.” We also thank the reviewer for raising the important question of whether a substantial portion of tasks in prior benchmarks—especially MMSearch—can be solved with text-only search. We conducted additional analyses to address this point, and we have added both empirical results and clarifications in the revised paper.
>
> **Text-Only Solvability in MMSearch-Plus**
>
> To directly estimate the degree to which tasks are solvable without genuine visual reasoning, we first implemented a text-only search baseline using the same agent framework but disabling all image retrieval tools. As shown in the updated Table 1, the model attains competitive accuracy in several categories solely through text search, confirming that a non-negligible portion can be solved without visual grounding.
>
> But this does not imply that MMSearch-Plus tasks do not enforce true multimodal reasoning. Instead, by manually inspecting agent trajectories, we observed that MLLMs usually transform visual elements into text queries. For example, logos in a sports event might be read by a model, converted to text, and used as a query in text search. As reviewer WMkC mentioned, scene text like logos can be seen as both visual and textual information. The model has (at least) two natural options:
>
> 1. Use its own OCR ability and call text search.
> 2. Use image search on the subregion containing text.
>
> Empirically, we found that for the o3 model, in 8 out of 10 cases, the model would choose the former - it converts visual elements to text for search.
>
> **Additional Observation Added to Section 5.1**
>
> The accuracies of o3 with text search (up to 20 rounds) and with full rollout are similar (37.0 vs. 36.0). However, their category-level accuracy distributions differ substantially. For example, image search yields better performance in Film/TV, where localizing subtle visual attributes is necessary, whereas additional rounds of text search provide greater benefits in Geography, which often requires retrieving factual external information rather than extracting fine-grained visual cues. This divergence supports our claim that both forms of retrieval are necessary and that MMSearch-Plus successfully stresses multimodal reasoning beyond what text search alone can provide.

---

> > ### Comment · Reviewer_xCaP · 2025-11-27
> >
> > Thanks a lot for the thoughtful rebuttal. A clarification question:
> > > we observed that MLLMs usually transform visual elements into text queries...The model has (at least) two natural options:
> > use its own OCR ability and call text search
> >
> > Does that mean multimodal search is not really required? This is the core motivation for the author's proposed benchmark.

---

> > > ### Author Response · Authors · 2025-12-02
> > >
> > > Thank you for the clarification question. Our observation that “MLLMs usually transform visual elements into text queries” primarily concerns **scene-text** cues (e.g., logos, signage, on-screen names). These are visual inputs, but they can often be converted into text with minimal loss of information—much like how humans typically search by *reading* a name and typing it, rather than cropping the name region and performing an image-based lookup.
> > >
> > > That said, this does **not** mean multimodal search is unnecessary. When we add a fine-grained **subregion image search** tool to the agent’s toolset, overall accuracy improves by **+0.6%**. While the aggregate gain appears modest, the **category-level changes** reveal where multimodal search is essential: in **Film/TV**, accuracy rises from **30% → 45%**, and **Geography** and **Tech** also improve by **+1.5%** and **+1.3%**, respectively. This indicates that image retrieval is disproportionately valuable in categories where success depends on localizing and matching subtle visual attributes that are difficult to express (or even perceive reliably) as text.
> > >
> > > In categories where performance decreases after adding image search (e.g., **Sports −5.6%**, **Vlog −5.4%**), we believe this is largely due to the model not always constructing effective image-search queries, and because some questions are better served by well-formed text queries with stronger web priors. Importantly, a capable agent can mitigate this issue by choosing *not* to invoke image search when it is unlikely to help—so these degradations reflect current tool-use policy limitations rather than evidence that multimodal search is unneeded.
> > >
> > > Finally, prior and concurrent work supports the importance of image retrieval in similar settings: **LiveVQA** [1] (with easier questions but comparable visuals) highlights the value of image search, and concurrent benchmarks such as **MM-BrowseComp** [2] and **BrowseComp-VL** [3] report consistent findings. Taken together—(i) scene-text can sometimes be textified, (ii) image search yields substantial gains in the categories that require fine-grained visual grounding, and (iii) related benchmarks corroborate the benefit—we conclude that **multimodal search remains essential** for fully solving MMSearch-Plus, even if a subset of tasks can be approached via OCR-to-text search.
> > >
> > > 1. LiveVQA: Live Visual Knowledge Seeking
> > > 2. MM-BrowseComp: A Comprehensive Benchmark for Multimodal Browsing Agents
> > > 3. WebWatcher: Breaking New Frontier of Vision-Language Deep Research Agent

---

### Author Response · Authors · 2025-12-03
**Rebuttal Summary**

We thank all reviewers for their thoughtful feedback and generally positive assessment of **MMSearch-Plus** as a benchmark for multimodal deep-research agents. We address the main concerns below.

### 1. Necessity of Multimodal Search

Additional experiments with a **text-only agent** show that some tasks can be partially solved without image tools. However, enabling **SoM-based subregion image search** yields consistent overall gains (+0.6%) and large improvements in visually demanding categories (e.g., Film/TV **30→45%**, Geography **+1.5%**, Tech **+1.3%**). Models often prefer OCR + text search, but **image search adds complementary visual signal** when nuances cannot be “textified.” Combined with trends from related benchmarks, this supports our claim that MMSearch-Plus **genuinely stresses multimodal reasoning**.

### 2. Novelty Beyond Prior VQA Benchmarks

MMSearch-Plus is **not** a web-augmented VQA dataset. It explicitly evaluates **agentic multimodal search**, requiring models to:

1. Select visual regions (SoM),
2. Choose between text vs. image tools,
3. Retrieve external evidence for tasks **unsolvable at construction time** without search.

Unlike OK-VQA/InfoSeek settings, where pretraining leakage is not uncommon, MMSearch-Plus emphasizes **provenance-aware retrieval under noise** and isolates the **research** component of deep reasoning.

### 3. SoM, Zooming, and Tool-Use Behavior

Tool use is optional and capped (20 calls). SoM provides **up to +3.9 point gains**, though current models mainly use zoom to refine internal visual understanding rather than chaining zoom → subregion search. Tool-usage patterns vary by model (e.g., Gemini: 12.8% zoom-then-search vs. o3: 2.4%). Inspection of Easy-split failures shows that drops often stem from **skipped image searches**, not over-retrieval or noise.

### 4. Dataset Size, Diversity, and Internal Knowledge

Although all final questions pass STE, the full pipeline includes curated videos, human-intuitive questions, and filtering to ensure **retrieval necessity** before difficulty elevation. To mitigate internal-knowledge shortcuts and limited size, we are:

* Releasing **key-region annotations** and **human trajectories**,
* Maintaining an **MMSearch-Plus-lite** subset (239 tasks unsolvable without search),
* Developing **semi-automated multimodal expansion** and periodic updates.

Short answers are intentional to focus evaluation on **research process**, not verbosity. Difficulty is better reflected by **trajectory complexity** (multi-hop, multi-tool reasoning). We will refine difficulty labels with human tool-use trajectories.

---

We hope these clarifications and new analyses address reviewer concerns and reinforce MMSearch-Plus as a rigorous, forward-looking benchmark for **agentic multimodal search**.

---

### Meta-Review · Area_Chair_AyCx · 2026-01-07

**Summary:**

The paper introduced a new multimodal searching benchmark to study whether multi-modal LLM agents can decide where to look at in an image and what tools to call before returning the answer. The data curation process effectively mitigates information leakage and discourages text-only solutions, and the proposed pipeline is sound. While some issues were raised regarding the size and diversity of the benchmark, computational cost, and the depth of analysis, the authors addressed most of these points during the discussion period. The reviewers are overall positive and the AC agrees. The AC therefore recommends acceptance and urges the authors to incorporate the reviewers’ feedback into the final version.

**Reviewer Concerns:**

See above.

**Reviewer Scores:**

See above.

---

### Decision · Program_Chairs · 2026-01-26

Accept (Poster)